# The differential impact of climate interventions along the political divide in 60 countries

Michael Berkebile-Weinberg ®[1,4], Danielle Goldwert ®[1,4], Kimberly C. Doell ®[2], Jay J. Van Bavel ®[1,3] & Madalina Vlasceanu[1] ✉

A major barrier to climate change mitigation is the political polarization of climate change beliefs. In a global experiment conducted in 60 countries (N = 51,224), we assess the differential impact of eleven climate interventions across the ideological divide. At baseline, we find political polarization of climate change beliefs and policy support globally, with people who reported being liberal believing and supporting climate policy more than those who reported being conservative (Cohen's $d = 0.35$ and 0.27, respectively). However, we find no evidence for a statistically significant difference between these groups in their engagement in a behavioral tree planting task. This conceptual-behavioral polarization incongruence results from self-identified conservatives acting despite not believing, rather than self-identified liberals not acting on their beliefs. We also find three interventions (emphasizing effective collective actions, writing a letter to a future generation member, and writing a letter from the future self) boost climate beliefs and policy support across the ideological spectrum, and one intervention (emphasizing scientific consensus) stimulates the climate action of people identifying as liberal. None of the interventions tested show evidence for a statistically significant boost in climate action for self-identified conservatives. We discuss implications for practitioners deploying targeted climate interventions.

Climate change is one of the most critical issues facing society, with the United Nations' Intergovernmental Panel on Climate Change urging rapid global decarbonization[1] as a necessary action to avoid irreversible ecological and societal loss[2]. Despite the broad scientific consensus identifying human activity as a significant contributor to this global crisis[3,4], beliefs about the reality of anthropogenic climate change and the extent to which climate change is a global emergency that warrants action have become increasingly politically polarized[5–7]. The political polarization of climate change is problematic given its detrimental impact on climate action and support for mitigation policies[8,9]. The current paper examines the political polarization of

climate change at the level of beliefs and behaviors, and the causal impact of several behavioral climate interventions on beliefs, policy support, and individual-level action across the ideological divide between people around the world identifying as liberal versus conservative.

A large body of literature has documented the robust link between political ideology and belief in climate change around the world[10]. For instance, a meta-analysis of 171 studies across 56 nations revealed that the strongest demographic correlate of climate change belief was political ideology, such that people who are liberal (or those who align with the political left) were more likely to believe in climate

[1]Department of Psychology, New York University, New York, NY, USA. [2]Department of Cognition, Emotion, and Methods in Psychology, Faculty of Psychology, University of Vienna, Vienna, Austria. [3]Norwegian School of Economics, Bergen, Norway. [4]These authors contributed equally: Michael Berkebile-Weinberg, Danielle Goldwert. ✉e-mail: vlasceanu@nyu.edu

change compared to people who are conservative (or those who align to the political right[11]). For example, centrist and left-wing party supporters and politicians in Australia had greater belief in anthropogenic climate change than right-wing supporters and politicians[12,13]. In the UK, greater levels of political conservatism predicted higher levels of climate change skepticism[14,15]. This pattern has also been documented in the US[16], where liberal-leaning individuals were more likely to accept the scientific consensus about anthropogenic climate change and express personal concern about global warming compared to conservative-leaning individuals[8]. Experiments have found that when their political identity was made salient, self-identified conservatives in Australia reported lower belief in anthropogenic climate change and were less likely to support climate change policies than self-identified conservatives whose identity was not made salient[17], suggesting that political identity has a causal influence on differences in climate change beliefs and policy support.

But is the political polarization of climate change belief around the world accompanied by a corresponding polarization of climate action? Previous literature supports two competing hypotheses. On the one hand, the political polarization observed at the level of belief in climate change could translate into behavioral polarization, by which climate change believers act to protect the environment while climate skeptics do not take such action (i.e., a belief-behavior polarization congruence). In support of such a green act hypothesis, a vast body of work has consistently found that beliefs are reliable predictors of behavior[18–22], even when it comes to ideological topics[23]. As such, there is good reason to believe that polarized beliefs will be mirrored in polarized action that can mitigate climate change (e.g., planting trees).

On the other hand, polarized beliefs about climate change might not correspond to an equivalent polarization of climate action (i.e., belief-behavior polarization incongruence). In support of such a green gap hypothesis, the predictive power of beliefs on behavior is moderated by a number of factors, including cognitive biases[24], perceptions of control[25] (for review, see ref. 18), personal costs[26], social norms[27–29], and efficacy beliefs[30]. For example, despite self-identified liberals' stronger beliefs in social equality compared to self-identified conservatives, the two ideological groups exhibited no differences in relevant behaviors (e.g., reducing inequalities around education, employment, housing) when these behaviors came at a personal cost[26]. In the climate change domain, work on consumer behavior has introduced the notion of a green gap to refer to the mismatch between consumers' pro-climate beliefs and their lack of sustainable behaviors in energy consumption, eating, and travel behaviors[31]. Such a gap between conceptual and behavioral signatures has been suggested to apply more strongly to those who report more pro-climate beliefs and values (i.e., liberal-leaning individuals); However, the mismatch between climate skeptics' beliefs and behaviors has also been documented, with farmers adopting pro-environmental practices despite lacking belief in anthropogenic climate change[32].

In addition to investigating the polarization in climate beliefs and behaviors, we sought to assess the differential impact of climate interventions across the ideological divide. Determining how different strategies impact partisans along the ideological spectrum is critical to create effective field interventions targeting different groups[33–35]. For example, political partisans tend to reject information that counters their preexisting beliefs[36]. Given pre-existing differences between the climate beliefs of people who are liberal and conservative[11], it could be that some climate interventions might be more effective on the former group. Indeed, climate skeptics–compared to climate believers–have been found to respond differently to climate interventions[37,38]. For example, scientific consensus messaging (i.e., informing the public that most scientists are in agreement about the climate crisis[39,40]) has had limited effects on climate skeptics' support for climate action[41–43], or has even sparked reactance and decreased support for climate policy[37,44]. Similarly, framing climate change in terms of moral

foundations has had differential impacts on partisans–while those identifying as liberal were not affected by such messaging, self-identified conservatives' pro-environmental attitudes and behaviors increased when climate change was framed in terms of binding moral foundations (e.g., loyalty to authority, purity[38]). Given these partisan differences in the responses to interventions, we assessed whether people who are liberal and conservative are differentially affected by a wide range of climate interventions.

In this work, we find political polarization of climate change beliefs and policy support around the globe, with people who reported being liberal believing and supporting climate policy more than those who reported being conservative. However, we find no evidence for a statistically significant difference between these groups in the degree to which they engaged in a behavioral tree planting task. We find that climate action on this task results from self-identified conservatives acting despite not believing in climate change rather than self-identified liberals failing to act on their beliefs. When assessing the effects of the interventions, we find that three boost climate beliefs and policy support across the ideological spectrum (emphasizing effective collective action, writing a letter to a future generation member, and writing a letter from the future self), and one intervention stimulates the climate action for those that identify as liberal (emphasizing scientific consensus).

## Results

We investigated whether engaging in climate action is ideologically polarized and whether partisans respond differently to climate interventions in a large sample spanning 60 countries ($N = 51,224$). We used a dataset collected as part of an international collaboration that empirically tested the efficacy of eleven climate action interventions (Table 1) compared to a no-intervention control condition at increasing climate change beliefs and behaviors[45]. The interventions tested were crowdsourced from behavioral scientists from around the world who answered a call for collaboration posted on professional societies listservs, forums, and on social media. The submitted interventions were screened for feasibility in an international context, relevance for the dependent variables, theoretical support from prior work, and other study-specific constraints (e.g., not involving deception, being administrable within 5 min, etc.). Subsequently, a sample of 188 behavioral science experts rated them on theoretical relevance and practical potential to climate mitigation. To implement the top interventions identified using this process, the researchers who proposed each intervention worked closely with researchers who had published theoretical work relevant to each intervention while seeking input from all 250 coauthors on the original study[45].

Importantly, several interventions were directly relevant to the issue of political polarization, given prior work suggesting they might differentially affect liberals and conservatives (e.g., scientific consensus[41]), and some were specifically developed based on theoretical work in political psychology (e.g., system justification[46], or binding moral foundations[38]).

Participants ($N = 51,224$, from 60 countries) were mostly recruited online or via convenience/snowball sampling. No statistical method was used to predetermine the sample size. Sample information by country can be found linked in the Supplemental Materials. After joining the study, participants were randomly assigned to one of the 11 interventions or a no-intervention control in which they read a passage from a literary text. Then, participants indicated their climate change beliefs, operationalized as their agreement (measured on a scale from 0 = Not at all to 100 = Extremely) with the following four statements (presented in a randomized order, $\alpha = 0.93$): Taking action to fight climate change is necessary to avoid a global catastrophe; Human activities are causing climate change; Climate change poses a serious threat to humanity; and Climate change is a global emergency. Participants also indicated their support for a set of nine climate mitigation

**Table 1 | Interventions, theoretical frameworks, and brief descriptions**[38,39,46,76–87]

| Intervention | Theoretical framework | Description |
| --- | --- | --- |
| Dynamic social norms | Sparkman and Walton (2017)[17] | Informs participants of how country-level norms are changing and "more and more people are becoming concerned about climate change", suggesting that people should take action. |
| Work together norm | Howe et al. (2021)[77] | Combines referencing a social norm (i.e., "a majority of people are taking steps to reduce their carbon footprint") with an invitation to "join in" and work together with fellow citizens toward this common goal. |
| Effective collective action | Goldenberg et al. (2018)[77]; Lizzio-Wilson et al. (2021)[79] | Features examples of successful collective action that have had meaningful effects on climate policies (e.g., protests) or have solved past global issues (e.g., the restoration of the ozone layer). |
| Psychological distance | Jones et al. (2017)[80] | Frames climate change as a proximal risk by using examples of recent natural disasters caused by climate change in each participants' nation and prompts them to write about the climate impacts on their community. |
| System justification | Feygina et al. (2010)[46] | Frames climate change as threatening to the way of life to each participant's nation, and makes an appeal to climate action, as the patriotic response. |
| Future-self continuity | Hershfield et al. (2012)[81] | Emphasizes the future self-continuity by asking each participant to project themselves into the future and write a letter addressed to themselves in the present, describing the actions they would have wanted to take regarding climate change. |
| Negative emotions | Chapman et al. (2017)[82] | Exposes participants to ecologically valid scientific facts regarding the impacts of climate change framed in a 'doom and gloom' style of messaging that were drawn from different real-world news and media sources. |
| Pluralistic ignorance | Geiger and Swim (2016)[83] | Presents real public opinion data collected by the United Nations that shows what percentage of people in each participant's country agree that climate change is a global emergency. |
| Letter to future generation | Shrum (2021)[87]; Wickersham et al. (2020)[84] | Emphasizes how one's current actions impact future generations by asking participants to write a letter to a socially close child who will read it in 25 years when they are an adult, describing current actions towards ensuring a habitable planet. |
| Binding moral foundations | Wolsko et al. (2016)[38] | Invokes authority (e.g., "From scientists to experts in the military, there is near universal agreement"), purity (e.g., keep our air, water, and land pure), and ingroup-loyalty (e.g., "it is the American solution") moral foundations. |
| Scientific consensus | van der Linden et al. (2015)[39], van der Linden (2021)[85]; Rode et al. (2021)[86] | Informs participants that "99% of expert climate scientists agree that the Earth is warming, and climate change is happening, mainly because of human activity". |

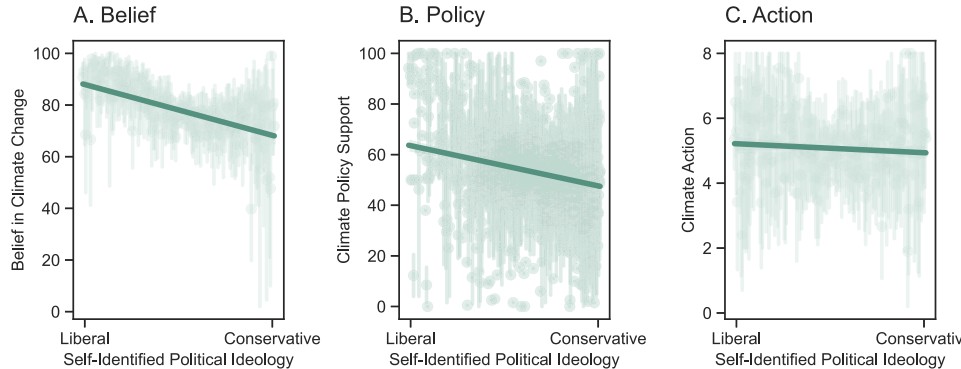

**Fig. 1 | Climate change belief, policy support, and action predicted by self-reported political ideology.** Belief in climate change (Panel **A**), climate policy support (Panel **B**), and climate action (Panel **C**), as a function of self-reported political ideology (measured from 0 = liberal to 100 = conservative), in the control condition ($N = 4302$). Vertical lines represent 95% bootstrapped confidence intervals of the means at each level of political ideology; Fitted lines represent the best fit linear regression lines.

policies (e.g., I support raising carbon taxes on gas/ fossil fuels/coal; see Supplement for the full list of policies; $\alpha = 0.88$). Finally, participants' climate action was assessed as their choice to opt into completing an optional, cognitively demanding tree-planting task (i.e., a modified version of the Work for Environmental Protection Task or WEPT[47]). The WEPT is a multi-trial, web-based procedure in which participants can choose to exert voluntary effort screening stimuli for specific numerical combinations (i.e., an even first digit and an odd second digit). Each completed page resulted in the actual planting of a tree through a donation to The Eden Reforestation Project. Therefore, participants had the opportunity to produce actual environmental benefits at actual behavioral costs, mimicking classic sustainable behavior tradeoffs[48,49]. Of note, given this incentivized behavioral task,

this project funded the planting of 333,333 real trees. Importantly, the WEPT has been validated and has been found to correlate with well-established scales for assessing pro-environmental behavior (e.g., General Ecological Behavior scale, GEB[50]) and with direct donation behaviors (e.g., the donation of a part of their payment to an environmental organization[47]). Participants then completed demographic variables scales (including their ideological leaning on social and economic issues), were debriefed, and compensated for their participation.

### Climate belief
To replicate prior patterns of political polarization of climate change beliefs in this large global sample, we first analyzed the participants in

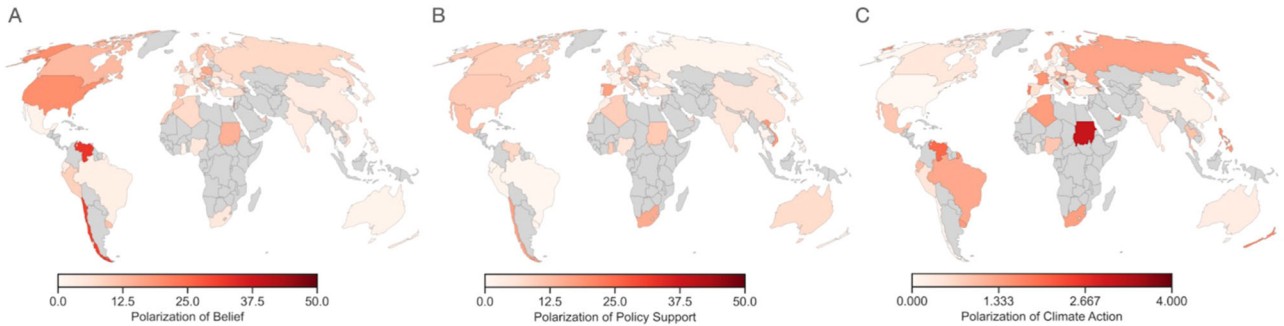

**Fig. 2 | Polarization of climate change belief, policy support, and action.** The degree of polarization (where higher scores reflect greater polarization), operationalized as the absolute value of the difference between self-reported liberals' and conservatives' climate beliefs (Panel **A**), policy support (Panel **B**), and action (Panel **C**), in the control condition (*N* = 4302). Each country's means and confidence intervals of these polarization scores for each outcome can be found in Figs. S1–S6.

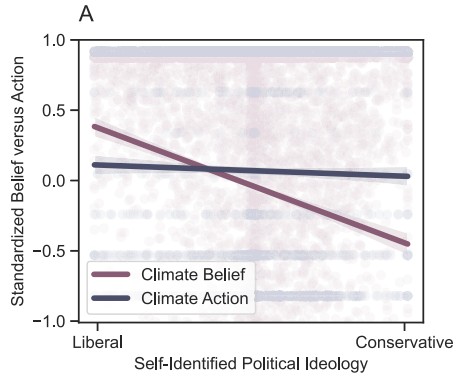
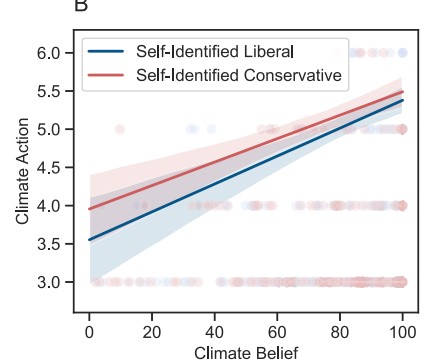

**Fig. 3 | Relations between self-reported political ideology, climate belief, and climate action in the control condition (*N* = 4302).** Panel **A**: Standardized scores of climate belief (in pink) and climate action (in gray) as a function of self-reported political ideology. Panel **B**: Mean climate action of self-reported liberals (in dark blue) and conservatives (in red) as a function of mean climate beliefs. For both panels, fitted lines represent the best-fit linear regression lines; Error bands represent 95% confidence intervals.

the control condition. For all analyses, the reported statistics follow from two-tailed tests. Political ideology was measured in a self-report fashion on a scale (ranging from 0 = liberal/left wing to 100 = conservative/right wing).

We first ran a linear mixed model with belief in climate change as the dependent variable, political ideology as the fixed effect, and by-item (i.e., 4 beliefs), by-participant, and by-country random effects. We found a significant effect of ideology, $\beta = -0.19$, SE = 0.02, $t(3714) = -11.47$, $p < 0.001$, $d = 0.35$, 95% CI = [−0.22, −0.16], such that the more liberal participants reported to be, the more they believed in climate change (Fig. 1A). This pattern held even after statistically adjusting for participants' age, gender, income, and education level, $\beta = -0.18$, SE = 0.02, $t(3702) = -10.79$, $p < 0.001$, $d = 0.35$, 95% CI = [−0.21, −0.15]. The effect also held, $\beta = -6.19$, SE = 0.77, $t(3714) = -8.04$, $p < 0.001$, $d = 0.26$, 95% CI = [−7.70, −4.68], when treating ideology as a binary variable (i.e., liberals versus conservatives) by taking a median split of the continuous ideology measure within each country ($M_{Liberal} = 80.7$, SD = 22.6; $M_{Conservative} = 74.2$, SD = 26.1; Fig S7A). This finding replicates the literature on the polarization of climate change beliefs (e.g., refs. 5,6,11) in a global sample spanning 60 countries (Figs. 2A; S1; S4).

## Climate policy support

We found similar effects of political ideology on support for climate change mitigation policy. We ran a linear mixed model with climate policy support as the dependent variable, political ideology (measured on a scale from 0 = liberal to 100 = conservative) as the fixed effect, and by-item (i.e., 9 policies), by-participant, and by-country random

effects. We found a significant effect of ideology, $\beta = -0.11$, SE = 0.01, $t(3695) = -8.18$, $p < 0.001$, $d = 0.27$, 95% CI = [−0.14, −0.08], such that the more liberal participants reported to be, the more they supported climate policy (Figs. 1B and 2B). This effect held, $\beta = -0.10$, SE = 0.01, $t(3678) = -7.65$, $p < 0.001$, $d = 0.25$, 95% CI = [−0.13, −0.08], even after statistically adjusting for participants' age, gender, income, and education level. The effect also held, $\beta = -3.33$, SE = 0.63, $t(3695) = -5.29$, $p < 0.001$, $d = 0.18$, 95% CI = [−4.56, −2.09], when treating ideology as a binary variable (i.e., liberals versus conservatives) by taking a median split of the continuous ideology measure within each country ($M_{Liberal} = 70.5$, SD = 18.9; $M_{Conservative} = 67.1$, SD = 20.5; Fig. S7B). The similar effects of ideology on beliefs and policy support are consistent with the strong relationship between the two constructs ($r = 0.68$, $p < 0.001$). The effect of ideology on both beliefs and policy support interacted with age, such that conservative ideology was associated with decreased belief and policy support, more so for older participants (Table S12).

But does the polarization of climate change at the conceptual level (i.e., beliefs and policy support) translate into polarization at the behavioral level (i.e., engaging in one type of individual-level climate action)?

## Climate action

To investigate whether the ideological polarization at the level of beliefs and policy support (i.e., conceptual level) also manifests at the level of individual behavior, we again focused on the data collected in the control condition. Given that the tree planting task (i.e., WEPT) was measured on an ordinal scale (i.e., 0 to 8 trees planted), we ran an

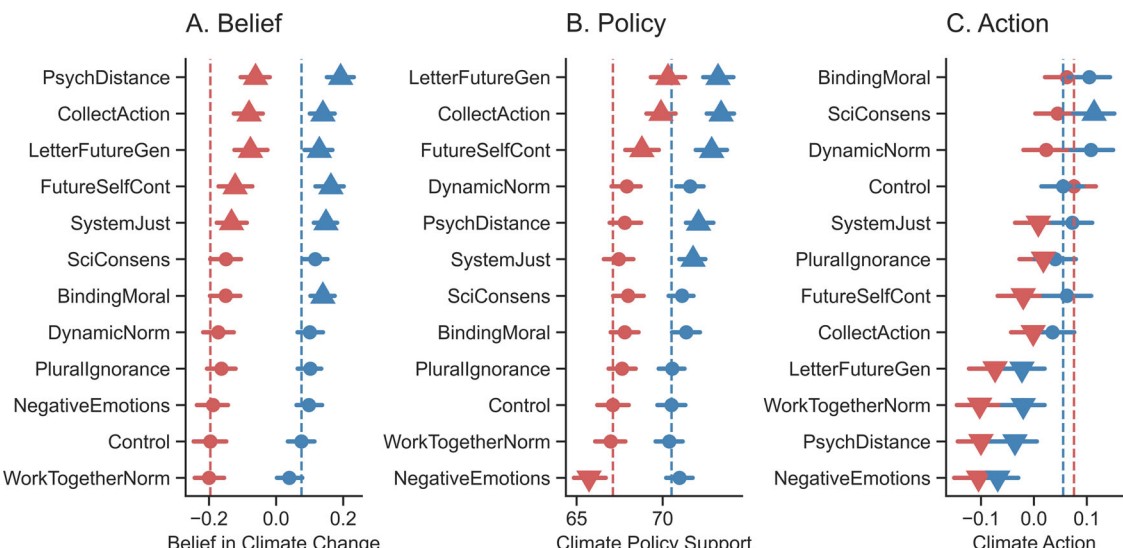

**Fig. 4 | Impact of each intervention on climate change beliefs, policy support, and action, separated by self-reported political ideology.** Interventions' impacts on climate beliefs (Panel **A**), policy support (Panel **B**), and action (Panel **C**), split by self-reported political ideology (liberals in blue and conservatives in red; $N = 51,224$). Vertical lines indicate the average in the control condition for each ideological grouping. Error bars represent 95% confidence intervals of the means. Upward triangles indicate significant increases, downward triangles indicate significant decreases, and circles indicate no statistically significant differences, always compared to the control.

ordinal mixed model (i.e., cumulative link mixed model fitted with the Laplace approximation) with the number of trees planted in the behavioral task as the dependent variable, political ideology as the fixed effect, and by-country random effects. We found no statistically significant evidence that participants' degree of climate action differed along the ideological spectrum, $\beta = -0.001$, SE = 0.001, $z(4214) = -0.621$, $p = 0.534$, $d < 0.001$, 95% CI = [−0.003, 0.002], (Figs. 1C and 2C). Given this null finding, we calculated a Bayes factor to quantify how likely the null hypothesis was compared to the alternative for this model. The Bayes factor for this model comparison was 0.043, suggesting that the null hypothesis (no effect of political ideology on climate action) was around 23 times more likely than the alternative hypothesis, thus lending strong support in favor of the null. As for climate change beliefs and policy endorsements, we found that the effects from the linear mixed effect analysis effect remained statistically non-significant, $\beta = -0.001$, SE = 0.001, $z(3798) = -0.574$, $p = 0.566$, $d < 0.001$, 95% CI = [−0.004, 0.002], when adjusting for participants' age, gender, income, and education level. The effect also remained statistically non-significant, $\beta = 0.05$, SE = 0.06, $z(4214) = 0.839$, $p = 0.402$, $d < 0.001$, 95% CI = [−0.16, 0.07], when treating ideology as a binary variable (i.e., liberals versus conservatives) by taking a median split of the continuous ideology measure within each country ($M_{Liberal} = 5.02$, SD = 3.40; $M_{Conservative} = 5.09$, SD = 3.43; Fig. S7C). Therefore, we found no statistical evidence that polarization of climate change beliefs and policy support translated into polarized individual-level behavior (Fig. 1C).

Furthermore, to quantify the interaction between belief and behavior as a function of political ideology, we transformed the belief ratings and the action ratings into standardized scores of each of the two types of outcome (i.e., belief and action). We then ran a linear mixed model with these standardized scores as the dependent variable, including a type (belief or action) by ideology (measured continuously from 0 = liberal to 100 = conservative) interaction as the fixed effect and by-participant and by-country random effects. We found a significant main effect of outcome type, $\beta = -0.274$, SE = 0.05, $t(4171) = -5.52$, $p < 0.001$, $d = 0.16$, 95% CI = [−0.37, −0.18], suggesting climate action was lower than climate beliefs (Fig. 3A). We also found a significant main effect of ideology, $\beta = -0.008$, SE = 0.0007, $t(8175) = -12.42$, $p < 0.001$, $d = 0.35$, 95% CI = [−0.01, −0.01], suggesting

that the more liberal participants reported to be, the more they believed in climate change and acted accordingly (Fig. 3A). Finally, we found a significant outcome type by ideology interaction, $\beta = 0.008$, SE = 0.0009, $t(4171) = 8.28$, $p < 0.001$, 95% CI = [0.006, 0.009], suggesting that the more conservative participants reported to be, the more their actions were stronger than their beliefs (Fig. 3A).

### Psychological process

This pattern of results poses a critical question regarding the underlying psychological process behind these effects: Is the differential impact of ideology on conceptual processes (i.e., belief and policy support polarization) compared to behaviors (i.e., no behavior polarization) driven by people who are liberal not acting on their beliefs (consistent with a liberal-oriented green gap), or could it be driven by people who are conservative acting despite their beliefs (consistent with a conservative-oriented green gap)? To investigate which of these two competing processes might be at play, we explored the degree to which beliefs predicted behaviors for each ideological group. In an ordinal mixed model with the number of trees planted in the behavioral task as the dependent variable, including political ideology (continuously measured) as it interacts with belief in climate change as the fixed effect, and by-country random effects, we found a significant main effect of ideology, $\beta = 0.011$, SE = 0.004, $z(4208) = 2.87$, $p = 0.004$, $d = 0.01$, 95% CI = [0.003, 0.02]. We also found a significant main effect of belief, $\beta = 0.017$, SE = 0.003, $z(4208) = 5.80$, $p < 0.001$, $d = 0.01$, 95% CI = [0.01, 0.02]. Finally, we found a significant ideology by belief interaction, $\beta = -0.0001$, SE = 0.00005, $z(4208) = -2.60$, $p = 0.009$, $d = 0.01$, 95% CI = [0.000002, 0.0002], such that the more conservative a participant reported to be, the less their beliefs about climate change predicted their climate behaviors (Fig. 3B; note that in this figure ideology is treated as binary for ease of a visual interpretation of the interaction). Policy support was also more strongly associated with the tree-planting behavior for self-identified liberals compared to self-identified conservatives (i.e., suggested by an ideology by policy interaction when predicting behavior: $\beta = -0.0002$, SE = 0.00006, $z(4195) = -2.97$, $p = 0.003$, $d = 0.01$, 95% CI = [−0.0003, 0.00008].

These results suggest that the polarization gap between belief and behavior is more likely explained by a belief-behavior incongruence in

| | Climate Beliefs | | Climate Policy Support | | Climate Action | |
|---|---|---|---|---|---|---|
| | Self-ID Liberals | Self-ID Conservatives | Self-ID Liberals | Self-ID Conservatives | Self-ID Liberals | Self-ID Conservatives |
| Psych. Distance | ✓ | ✓ | ✓ | | ✗ | ✗ |
| Letter to Future Gen. | ✓ | ✓ | ✓ | ✓ | ✗ | ✗ |
| Collective Action | ✓ | ✓ | ✓ | ✓ | | ✗ |
| Future-Self Cont. | ✓ | ✓ | ✓ | ✓ | | ✗ |
| Neg. Emotions | | | | ✗ | ✗ | ✗ |
| System Justification | ✓ | ✓ | ✓ | | | ✗ |
| Sci. Consensus | | | | | ✓ | |
| Binding Moral Found. | ✓ | | | | | |
| Dynamic Norms | | | | | | |
| Pluralistic Ignorance | | | | | | ✗ |
| Work Together Norm | | | | | ✗ | ✗ |

**Fig. 5 | Interventions' effects on self-reported liberals' and conservatives' (N = 51,224) climate beliefs, policy support, and action, compared to the control condition.** Green check marks indicate significant increases compared to control, red X marks indicate significant decreases compared to control, and empty cells indicate no statistically significant differences compared to control. The coefficients of these analyses can be found in Tables S1–S6.

people who are conservative rather than people who are liberal. That is, instead of self-identified liberals not acting on their beliefs, the data are more consistent with self-identified conservatives acting despite their beliefs, in line with prior work on the belief-behavior disconnect in Republicans but not Democrats in the United States[23].

This interpretation is particularly promising for interventions aiming at increasing climate action, a critical component of the climate crisis response. Our data suggest that the political resistance to believing in climate change or supporting climate policy may not translate into a behavioral resistance to engaging in at least one type of individual-level climate action. To test this hypothesis, we investigated the impact of 11 climate interventions on the climate beliefs, climate policy support, and individual-level climate action of self-identified liberals and conservatives.

## Interventions' impact on beliefs
Next, we investigated the impact of the 11 climate interventions (Table 1) on self-identified liberals' and conservatives' climate beliefs. We ran a linear mixed effects model with belief in climate change as the dependent variable, condition (11 interventions versus control) as it interacts with political ideology (median split within each country) as the fixed effects, including by-participant and by-country random effects. Across our entire sample of participants ($N = 51,224$), we found a main effect of political ideology, $\beta = 6.35$, SE = 0.71, $t(49721) = 8.89$, $p < 0.001$, $d = 0.29$, 95% CI = [4.95, 7.75] (Fig. 4A; Table S7). We also found the main effects of the condition but no condition-by-ideology interactions. Thus, we found no statistically significant evidence that those who identified as liberals or conservatives were, on average, differentially affected by climate interventions relative to the control condition (Fig. 4A; Table S7).

To determine the impact of different interventions within each ideological group, we also ran linear mixed models separately for the liberals and conservatives. Each model included belief in climate change as the dependent variable, condition as the fixed effect, including by-participant, by-item (4 beliefs), and by-country random intercepts. For people who are liberal, we found that six interventions significantly increased climate beliefs (decreasing psychological distance, emphasizing effective collective actions, future-self continuity, writing a letter to the future generation, system justifying messaging, and binding moral foundations; Table S1; Figs. 4 and 5). Five of these interventions also increased the climate beliefs of self-identified conservatives (decreasing psychological distance, emphasizing effective collective actions, future-self continuity, writing a letter to the future generation, and system justifying messaging; Table S2; Figs. 4 and 5). Thus, five of the eleven interventions we tested were effective at increasing beliefs about climate change across the ideological divide (Figs. 4 and 5).

## Interventions' impact on policy support
Next, we investigated the impact of the eleven climate interventions (Table 1) on self-identified liberals' and conservatives' climate policy support. We ran a linear mixed effects model with climate policy support as the dependent variable, condition (11 interventions versus control) as it interacts with political ideology (median split within each country) as the fixed effects, including by-participant and by-country random effects. We found a main effect of political ideology, $\beta = 3.36$, SE = 0.58, $t(50280) = 5.80$, $p < 0.001$, $d = 0.05$, 95% CI = [2.25, 4.50] (Fig. 4B; Table S8). We also found main effects of the condition, and a condition by ideology interaction for the negative emotion intervention, $\beta = 1.74$, SE = 0.81, $t(50260) = 2.13$, $p = 0.033$, $d = 0.02$, 95% CI = [0.15, 3.33], suggesting that this intervention decreased the policy support of self-identified conservatives more than liberals (Fig. 4B; Table S8).

To determine the impact of different interventions on the climate policy support of each ideological group, we ran linear mixed models separately for people who are liberal and conservative. Climate policy support was included as the dependent variable, condition as the fixed effect, including by-participant, by-item (9 policies), and by-country random intercepts. We found that, compared to the control condition, five interventions significantly increased self-identified liberals' climate policy support (emphasizing effective collective actions, writing a letter to the future generation, future-self continuity, decreasing

psychological distance, and system justifying messaging; Table S3). Three of these interventions increased self-identified conservatives' climate policy support (emphasizing effective collective actions, writing a letter to the future generation, and future-self continuity; Table S4), and one intervention backfired (negative emotion messaging; Table S4). Thus, three of the eleven interventions (emphasizing effective collective actions, writing a letter to the future generation, and future-self continuity) were effective at increasing support for climate mitigation policy across the ideological divide (Fig. 5).

#### Interventions' impact on behaviors

To investigate the impact of the 11 climate interventions on self-identified liberals' and conservatives' climate action (engagement in the tree planting task), we ran an ordinal mixed model (i.e., cumulative link mixed model fitted with the Laplace approximation) with the number of trees planted in the behavioral task as the dependent variable, condition (11 interventions versus control) as it interacts with political ideology (median split within each country) as the fixed effects, including by-country random effects. We found no statistically significant main effect of political ideology, $\beta = -0.04$, SE $= 0.06$, $z(51186) = -0.70$, $p = 0.480$, $d < 0.001$, 95% CI $= [-0.16, 0.07]$, several significant effects of condition (Table S9), and a significant condition by ideology interaction for six out of the 11 interventions (Table S9), in each case suggesting that the intervention decreased the tree planting efforts of people who are conservative to a higher extent (Fig. 4C, Table S9). These effects held when statistically adjusting for time participants spent on the intervention phase (Table S10).

To determine the interplay between the interventions' impact and their length on the tree planting outcome, we also ran an ordinal mixed model with the number of trees planted as the dependent variable, condition as it interacts with intervention time as the fixed effects, including by-country random effects (Table S11). On average, spending more time on the intervention phase predicted more trees being planted. However, this average effect manifested differently across interventions. Specifically, spending more time on the intervention phase increased the number of trees planted in the scientific consensus and binding moral foundations interventions, did not have a statistically significant impact on tree planting in the dynamic norms intervention, and decreased the number of trees planted in all the other 8 interventions (Table S11). These results suggest that a reason for the negative effects observed in some of the interventions might have been due to the limited time budget participants had for the study, such that less time was allocated to planting trees in the conditions with longer interventions. The results may also suggest that in the absence of time constraints, some interventions (e.g., scientific consensus and binding moral foundations) might even increase individual-level pro-environmental behavior. However, the degree to which these findings generalize to behaviors that do not hinge on time (e.g., direct donations) should still be empirically assessed in future studies.

To further investigate the impact each intervention had on self-identified liberals' climate behavior, we ran an ordinal mixed model with climate action as the dependent variable and condition as the fixed effect, including by-country random intercepts. Compared to the control condition, one intervention significantly increased the climate action of people who identified as liberal (scientific consensus; Table S5), and four interventions significantly backfired (letter to the future generation, negative emotions, decreasing psychological distance, and working-together norms; Fig. 5; Table S5). For people who identified as conservative, eight of the eleven interventions significantly backfired (emphasizing effective collective actions, future-self continuity, writing a letter to the future generation, negative emotions, pluralistic ignorance, decreasing psychological distance, system justifying messaging, and working-together norms; Fig. 5; Table S6). Thus, while self-identified liberals' climate action was

stimulated by one of the eleven interventions (scientific consensus), four interventions significantly decreased the tree-planting efforts of people across the ideological divide (letter to the future generation, negative emotions, decreasing psychological distance, and working-together norms; Fig. 5).

## Discussion

In a global study spanning 60 countries, we assessed the political polarization of climate change beliefs, climate policy support, and individual-level climate action, as well as the effectiveness of eleven climate interventions at increasing these three climate mitigation outcomes across the ideological divide. Replicating prior work[5,6,11,51], we found a consistent relationship between political ideology and climate beliefs and policy support, whereby people who identify as liberal believe in the threatening nature of anthropogenic climate change more than people who identify as conservatives. However, we found no evidence for statistically significant differences in the number of trees planted by people along the ideological spectrum. When assessing whether the conceptual-behavioral disconnect observed was triggered by people who are liberal not acting on their beliefs (e.g., a liberal-oriented green gap) or people who are conservative acting despite their beliefs (e.g., a conservative-oriented green gap), we found that self-identified conservatives' beliefs predicted their behaviors less than self-identified liberals' beliefs predicted their behaviors. This suggests that the disconnect between beliefs and behaviors in these results could be more strongly driven by participants who identify as conservative acting in a more pro-environmental manner than their beliefs would predict. This result aligns with recent findings in the United States[23], and is more consistent with a process by which participants who are conservative contributing to tree planting efforts despite not believing in the urgency of climate change as much as participants who are liberal[52–54]. Instead, they may have acted in this pro-climate way for reasons other than climate concern, such as to preserve and protect traditional values of nature and purity[55]. Alternatively, participants who identified as conservative may have conceptualized tree planting as an alternative to system-level action on climate. Future research should disentangle these processes, which could help assess the generalizability of these findings to other behaviors, both at the individual as well as at the collective or systemic levels.

We also found ideological differences in the impact of climate change interventions on climate beliefs, policy support, and individual-level action. Critically, these ideological effects differed across the three dependent variables, consistent with prior work pointing to the importance of the outcome when assessing the effectiveness of climate interventions[56]. Six of the eleven tested interventions increased the climate beliefs of participants who identified as liberal (decreasing psychological distance, writing a letter to a future generation, emphasizing effective collective actions, future-self continuity, system justifying messaging, and binding moral foundations); five interventions increased their policy support (emphasizing effective collective actions, writing a letter to the future generation, future-self continuity, decreasing psychological distance, and system justifying messaging); and one intervention increased their climate action (scientific consensus).

For participants who identified as conservative, five interventions increased their climate beliefs (decreasing psychological distance, emphasizing effective collective actions, future-self continuity, writing a letter to the future generation, and system justifying messaging), three interventions increased their policy support (writing a letter to the future generation, emphasizing effective collective actions, and future-self continuity), but no intervention increased their climate action. Instead, eight interventions significantly decreased their tree planting efforts (emphasizing effective collective actions, future-self continuity, writing a letter to the future generation, negative emotions,

pluralistic ignorance, decreasing psychological distance, system justifying messaging, and working-together norms).

These findings paint an optimistic picture for practitioners such as policymakers and climate communicators interested in increasing global beliefs in the severe threat posed by anthropogenic climate change and support for mitigative policies. Several interventions (e.g., writing a letter to the future generation or emphasizing effective collective action) were effective at boosting these conceptual processes across the ideological divide. However, when it comes to stimulating individual-level climate action, our findings suggest that practitioners around the world could successfully deploy scientific consensus messaging, but only when targeting people who are liberal. This finding aligns with prior work suggesting that scientific consensus messaging has limited effects on climate skeptics' support for climate action[41–43]. When targeting the climate action of people who are conservative, the behavioral toolbox for interventions in this space is sparser. Accordingly, the behavioral science field would benefit from future research investigating intervention strategies aimed at stimulating climate action across ideological divides.

Given the importance of political polarization in addressing climate change, these findings also advance theorizing. First, we provide additional explanations for the green gap phenomenon, previously discussed as liberals' failure to act on their pro-environmental beliefs. Here, we find the green gap can also arise through the converse process—climate actions (e.g., planting trees) can be elicited in people who are conservative in spite of their climate change skepticism. Additional evidence for eliciting climate action without attempting to change beliefs comes from the interventions' effects on the actions of participants who identified as conservative. That is, most interventions decreased the number of trees they planted, suggesting that when framed as climate change solutions, people who are conservative engage in pro-environmental behaviors to a lesser extent. Thus, in future work, we are interested in exploring alternative processes for eliciting pro-environmental behaviors that don't involve changing climate beliefs.

Second, our study establishes important boundaries on several prominent psychological theories. For instance, norm-based theories (e.g., dynamic norms, pluralistic ignorance, work-together norm), previously considered state-of-the-art in designing climate interventions[57], did not significantly increase climate beliefs, policy support, or individual action in this global sample. This is likely due to the diversity of our sample and the large heterogeneity of effects between countries. For example, correcting pluralistic ignorance increased climate beliefs by 5% in the US and Denmark but decreased beliefs by 7% in Romania and by 5% in India. Since many theories are established in WEIRD countries[58,59], our research suggests that these theories might not apply outside these contexts. As such, these findings suggest there is a serious need to develop and test theories of climate belief and action across cultures. For a rapid assessment of these interventions' effects across each of the four outcomes and across a range of variables (i.e., including country, political ideology, gender, age, socioeconomic status, income, and education), we created a web tool https://climate-interventions.shinyapps.io/climate-interventions/. We hope this data exploration resource can facilitate the advancement of science by offering researchers the ability to test additional hypotheses, which should then be empirically verified in follow-up experiments. We also urge scholars to incorporate these findings into their theories.

A critical component of our design that must be considered when interpreting these findings is the operationalization of climate action as the number of pages completed in the tree planting task. While this assessment of pro-climate behavior allowed a standardized measure of action across the 60 nations in which the experiment was deployed, it is limited to a single type of private mitigation behavior that offers a highly individual-focused solution to combating climate change. Therefore, these findings might not generalize to collective or

systemic climate actions, also critical to climate mitigation[60]. In future research, scholars should study additional pro-climate behaviors, especially ones that build towards collective solutions to this fundamentally collective problem[61], such as advocacy[62] or voting[63].

Another limitation of the current study is the attrition rate observed (36.4%) between the number of participants who completed the study and the participants whose data ended up in the final analysis. Although within the expected range of attrition for online studies (30–50%[64]), this feature of our data should be considered when interpreting the results. Future research should thus replicate the current findings in more controlled environments that may benefit from lower attrition rates.

Notably, the interventions tested here were homogeneously administered and not ideologically tailored. Given recent work showing that targeted interventions can be up to 200% more effective[65], we hope our results can inform and increase the potential of future targeted interventions in this space. Moreover, given the heterogeneity of the effects observed across ideologies and outcomes, we also recommend future research targeting mechanisms to consider these dimensions for optimal impact.

Overall, the present work provides global evidence of the complex relationship between political ideology and climate change beliefs, policy support, and individual climate action, providing evidence for a conservative-oriented green gap. Our analyses also have critical implications for the design and deployment of theoretically derived targeted interventions aimed at boosting climate awareness and action around the world. Scholars and policymakers in this space can leverage our findings to implement interventions selected to optimize outcomes, given the ideological composition of a target community, for a more effective, empirically informed response to the climate crisis.

## Methods
### Ethics approval
Ethics approval was obtained independently by each research team from the respective Institutional Review Board (IRB) associated with their institution. Analyses only included datasets submitted along with IRB approval.

### Participants
Participants' data came from a previously collected dataset[45]. A total of 83,927 completed the study between July 2022 and July 2023. Of them, 59,440 participants from 63 countries passed the two attention checks (i.e., Please select the color purple from the list below; To indicate you are reading this paragraph, please type the word 60 in the text box below.) and correctly complete the WEPT demo. Of this sample, ideological information was only available for 51,224 participants ($M_{age} = 39.62$, $SD_{age} = 15.82$; 29,410 women, 27,232 men, 400 nonbinary or other gender, 2398 declined to state gender) from 60 countries. The control condition contained a sample of 4,302 participants ($M_{age} = 39.52$, $SD_{age} = 15.64$; 2207 women, 2041 men, 35 nonbinary/other gender, 19 declined to state gender). Gender was collected via self-report and was included (along with age, education, and income) as a covariate in the primary analyses reported here to assess for potential effects on the measured outcomes. Given the differences in data collection timelines in each country, the initial version of this manuscript did not include this full sample; upon data collection completion in July 2023, we added data from additional countries, resulting in this final dataset. The results between the initial partial sample and the final sample did not differ.

### Experimental design and measures
Participants recruited for the study were asked to first read and sign informed consent. They were then exposed to the first attention check (Please select the color purple from the list below. We would like to

make sure that you are reading these questions carefully.). Following this, participants were given a definition of climate change: Climate change is the phenomenon describing the fact that the world's average temperature has been increasing over the past 150 years and will likely be increasing more in the future. After reading this definition, participants were randomly assigned to one of 12 conditions: 11 experimental interventions (Table 1) or a no-intervention control condition in a between-subjects design. Participants in the control condition were asked to read a short, thematically unrelated text from the novel Great Expectations by Charles Dickens, while participants in the experimental conditions were exposed to an intervention. Subsequently, participants in all conditions were asked to rate (in random order) their (1) climate beliefs, (2) climate policy support, and (3) willingness to share climate information on social media. Finally, participants were given the chance to contribute to tree-planting efforts by completing the WEPT. Then, participants in the control condition were asked to complete an additional set of variables. Finally, all participants were asked to fill out a series of demographics, which included another attention check (In the previous section, you viewed some information about climate change. To indicate you are reading this paragraph, please type the word sixty in the text box below.). We administered the entire survey in the participants' primary language of their country of residence.

## Outcome variables

**Climate beliefs.** Climate beliefs were operationalized as participants' responses to the question: How accurate do you think these statements are? (from 0 = Not at all accurate to 100 = Extremely accurate). The four statements were: Taking action to fight climate change is necessary to avoid a global catastrophe; Human activities are causing climate change; Climate change poses a serious threat to humanity; and Climate change is a global emergency. Cronbach's alpha of this scale in this dataset was 0.866.

**Climate policy support.** Climate policy support was measured as participants' level of agreement (from 0 = Not at all to 100 = Very much so), with the following nine statements: I support raising carbon taxes on gas/fossil fuels/coal; I support significantly expanding infrastructure for public transportation; I support increasing the number of charging stations for electric vehicles; I support increasing the use of sustainable energy such as wind and solar energy; I support increasing taxes on airline companies to offset carbon emissions; I support protecting forested and land areas; I support investing more in green jobs and businesses; I support introducing laws to keep waterways and oceans clean; I support increasing taxes on carbon intense foods (for example meat and dairy). Cronbach's alpha of this scale in this dataset was 0.844.

**WEPT Tree planting efforts.** To measure action with real-world impact performed at a cost to participants, we used a modified version of the work for environmental protection task (WEPT). This task is a multi-trial online procedure that detects consequential pro-environmental behavior by allowing participants the opportunity of engaging in voluntary cognitive effort (i.e., screen numerical stimuli) in exchange for donations to an environmental organization. This measure has been validated and has been found to correlate to self-reports and objective observations of other pro-environmental behaviors and conceptually related measures. Participants were first exposed to a demonstration of the WEPT, in which they were instructed to identify all target numbers for which the first digit is even and the second digit is odd (4 out of 18 numbers for the demonstration). Participants could only advance to the next page upon correctly completing the demonstration. There, they were told that planting trees is one of the best ways to combat climate change and that they would have the opportunity to plant up to 8 trees if they chose to engage in additional pages of the task (one tree per page completed). These pages

contained 60 numbers per page, which participants had to screen for target numbers. Alongside these instructions, participants were shown a pictogram of 8 trees, one of which was colored green to mark their progress in the task. Participants could exit the task at any point without penalty.

**Social media sharing intention.** Participants were first presented with the text: Did you know that removing meat and dairy for only two out of three meals per day could decrease food-related carbon emissions by 60%? It is an easy way to fight #ClimateChange #ManyLabsClimate \${e://Field/cond} source: https://econ.st/3qjvOnn (where {e://Field/cond} was replaced with the condition code for each group). Participants were then asked: Are you willing to share this information on your social media? The answer options were: Yes, I am willing to share this information; I am not willing to share this information; I do not use social media. The present investigation does not focus on this intention measure; however, the results are reported in the supplement for completion (Tables S13–S15).

## Demographic variables

Participants were asked to indicate their gender (male, female, non-binary/other, prefer not to say), age (in years), education level (in years of formal education completed), household income, and political orientation for economic and social issues (on two scales ranging from 0 = Extremely liberal/Left-wing to 100 = Extremely conservative/Right wing). To create an aggregated ideological leaning measure, we took the average of participants' answers on the two political orientation questions and treated that as the continuous measure of ideology (from liberal to conservative). We aggregated these two political ideology measures given a robust positive correlation between the two items ($r = 0.71$, $p < 0.001$), a prevalent procedure in this literature[66,67]. For a binary version of this continuous measure, we computed a median split of the continuous ideology measure within each country and labeled participants who scored above their country's ideology average as conservatives and participants who scored below their country's ideology average as liberals.

## Ethics and inclusion statement

This research utilized data collected by the International Collaboration to Understand Climate Action. Local researchers from across the world (60 countries in total represented in the data in this research) were invited to collaborate on the design, distribution, and analysis of the study, as well as lead individual research projects stemming from the collected data. Collaborator roles and responsibilities were agreed to before data collection began. For each research team in each country, individual IRB approval by local ethics review committees was required before data collection began. For the full list of IRB review committees, refer to the Supplement.

## Statistical methods

Throughout the "Results" section, we reported hierarchical mixed effects models to assess the effects of interest. Of note, these multi-level models were used to account for the data being non-independent within countries, participants and items (for beliefs and policy support) and have the benefit of alleviating multiple comparison concerns by performing partial polling in generating estimates[68]. All linear mixed models were run in R[69] with the lme4 package[70]. Cumulative link mixed models were run with the ordinal R package[71]. Bayes Factors were calculated using the BayesFactor R package[72]. Although the mixed effects models used here are largely robust to distributional assumption violations (see ref. 73), we conducted robust mixed effect models and weighted least squares mixed models to complement our primary analyses and formally account for any potential violations of residual normality and homoscedasticity, respectively. For each set of analyses, we found identical results to those reported in the main text.

Therefore, we are highly confident in the robustness of our models and results.

## Reporting summary
Further information on research design is available in the Nature Portfolio Reporting Summary linked to this article.

## Data availability
The dataset can be openly accessed on OSF: 10.17605/OSF.IO/YTF89[74].

## Code availability
The analysis scripts can be accessed on Zenodo: 10.5281/zenodo.10815267[75].

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

## Acknowledgements

This work was supported by the following grant funding: Google Jigsaw grant (M.V., K.D., J.J.V.B.); Swiss National Science Foundation, P400PS_190997 (K.D.); Templeton World Charity Foundation, TWCF-2022-30561 (J.J.V.B.; doi.org/10.54224/30561); NYU Climate Change Initiative Seed Grants (J.J.V.B., M.V., K.D.). The authors thank the International Collaboration to Understand Climate Action for the data collection efforts: https://manylabsclimate.wordpress.com/.

## Author contributions

M.B., D.G., J.J.V.B., and M.V. contributed to the conception or design of the work. Data acquisition was completed by K.C.D., J.J.V.B., and M.V. M.B. and M.V. analyzed and interpreted the data. The original draft was written by M.B., D.G., and M.V. Revisions were completed by M.B., D.G., K.C.D., J.J.V.B., and M.V.

## Competing interests

The authors declare no competing interests.
