## [Peer Review File · Nature Communications]

The Differential Impact of Climate Interventions along the Political Divide in 60 CountriesReviewer #1 (Remarks to the Author):

I found myself a little conflicted reading this paper. On the one hand: what a great dataset, a tribute to the authors' leadership and organizational skills! Also, I liked the big-picture framing, and the paper is presented professionally and clearly (minor asides: a critical number is missing in par. 2 of section 2.2 and in my copy I can't see the promised vertical dotted lines in Fig 4.)

On the downside, I'm not sure I feel confident in the two headline conclusions: (1) that differences in political ideology disappear when it comes to costly behavior and (2) that climate change interventions backfire for conservatives. Let me take each of these separately.

The paper shows that differences in political ideology – robust when examining climate beliefs and policy support – disappear when it comes to climate action. This is an attention-grabbing conclusion, and is probably the most memorable part of the paper. The takehome for many readers will be that liberals are basically virtue signalling around climate change. People paying closer attention will see that this isn't quite the message: that for whatever reasons conservatives are acting more green than their belief systems would imply.

My main hesitation here is around the operationalisation of climate action as tree-planting. Tree-planting is well-understood to be conservatives' favorite climate action because it doesn't ask people to accept structural changes in how society and business is structured. What conservatives rail against is government regulation that constrains the freedoms of individuals and markets. Climate action that disrupts the status quo, implies lifestyle changes, and challenges the hegemony of fossil fuel vested interests would probably see robust ideological divides. Individual, banded solution that do not disrupt the status quo and have broad green benefits beyond climate change (e.g., greening public spaces) would see smaller ideological divides. A revised paper should carry that caveat and perhaps soften the conclusion accordingly.

The second conclusion is that climate change interventions are largely ineffective with regard to behavior and potentially backfire, especially for conservatives (i.e., psychological interventions mostly don't work and potentially do harm). Here, I worry that the design carries a confound that could help explain the pattern (I should emphasise this criticism applies only to the behavior measure). WEPT requires participants to put in extra effort and time over and above the core business of reading the materials carried in the stimuli and responding to the DVs. The more time-consuming or cognitively taxing the intervention, one might expect that participants will have fewer resources to spare on the WEPT task. In other words, the design may be biased toward showing greater action in conditions that are easy and quick. It's difficult to know how big a problem this is on the basis of the information provided. The control condition required people to read an extract from Dickens. If the intervention tasks were more onerous than this, it might explain the high level of backfiring observed. I wonder if the authors have access to time data that can help referee whether participants spent as much time on this task as they did on the 11 interventions?

Gazing the description of the interventions, some of them almost certainly required more time / investment than others. Psychological distance, future-self continuity, and letter to future generation interventions required participants to do a thinking and writing task, which is much more involved in terms of time and effort than simply reading materials. Eyeballing Fig 4 reinforces my suspicion that there may be a confound here. For example, psychological distance is the MOST effective intervention on beliefs, effective for both liberals and conservatives, but the LEAST effective on behavior, apparently backfiring for both liberals and conservatives. Similarly, the letter to future generations task is the MOST effective intervention on policy support, but the FOURTH LEAST effective on behavior. Either there is something incredibly strange going on here psychologically, or participants were simply exhausted by these interventions and didn't have time for the WEPT.

In conclusion, I'm definitely intrigued by the data and open-minded to both takehome messages. But

these messages are combusive ones and so the authors would want to make sure they are robust and that the paper carries all the necessary nuance.

Three final points of clarification. First, it looks like this is a secondary analysis separate from another manuscript currently under review but drawing on the same dataset. It would be useful to know what that other paper was about so reviewers and editors can judge the scale of conceptual overlap.

Second, it looks like there was massive attrition (>80k respondents whittle down to <55k usable participants). Any thoughts on why, and what this means for the integrity of the dataset?

Third, what does it mean to say that 69% of countries included representative samples on at least one demographic variable? It might be better to just accept that these are convenience samples rather than straining too hard to make a case for representativeness.

Reviewer #2 (Remarks to the Author):

This paper reports the results of a mega-study on climate change interventions. It also makes very good use of the large control group to arrive at some important insights about the correlates of beliefs, policy support, and behavior. There are several crucial lessons, perhaps most importantly about the need to carefully differentiate outcome variables – both when looking at correlates and considering interventions. I was aware of this project, and it was a pleasure to read the paper. I support publication and have just a few suggestions for the authors to consider. First, there is some other literature that has shown differences across outcome variables that may be worth considering. This is summarized more less in: https://www.tandfonline.com/doi/full/10.1080/17524032.2020.1805343?casa_token=gL28KTP96aIAA AAA%3AuMIPGucbAFFI9sOBrc6YI8En2R0cHrmS-mKkSjbsSO2C3IsDI3MqInpYM4Opjf223kCzeFcW9Z0GCNY. Put another way, prior work also had shown distinct effects on outcomes (also see, e.g., Key and Campbell 2014). This study is a better demonstration, but it may be worth connecting it to that prior discussion. Second, the other literature which is worth mentioning is the debate on backfiring effects between van der Linden and Kahan—the findings here raise interesting questions about the psychological processes at work. Third, I would be interested in whether the “taking action...” belief variable demonstrated any distinct dynamics from the other three items in that composite given it involves action and is in fact fairly similar to what the gateway belief model uses for climate action. This may be worth a footnote (given the tree planting results). Fourth, did the authors check an interaction with age (and ideology) in the correlational results as some prior work suggests younger conservatives are more supportive of climate policies and actions (e.g., see Pew polling). Fifth, I found the discussion around Figure 3 a bit too sparse. Sixth, I really liked the exploration of the relationship between beliefs and behaviors in terms of whether liberals are not acting or conservative acting despite beliefs. Can the authors do the same thing with policy—as a precursor to behavior and a consequence of belief (particularly the former)? Even if there is no relationship in that case, it would be interesting to me. Seventh, my understanding is that none of the interventions allowed for targeted messages for liberals and conservatives (all were homogenous treatments)? If that is correct, I would make it clearer. Along those lines, my most salient point is that I think one implication of the results is that climate communicators need to consider generalized versus targeted messages across three distinct outcomes. That results in a 2x3 framework that probably should be explored in future work and scholars need to stop bundling messages and outcomes without being explicit. Finally, a small point – given pre-registration and in essence this is 11 distinct experiments, I do not think multiple comparisons are a problem, but the authors may want to mention it in the methods section to address the point as others may think of it.

Reviewer #3 (Remarks to the Author):

This appears to be the second paper to be published based on a herculean data collection effort in 60 countries. The first one (Reference 45 in the current manuscript and also attached for the reviewers' benefit, with an author count of about 300) reports on the effectiveness of 11 interventions to increase climate concern and action, crowdsourced by experts and tested in a between-subject design, on four DVs: belief in climate change, support for climate policy, willingness to share climate information on the internet, and a costly personal behavior. These results are further unpacked for respondents who differ in their initial belief in climate change, and country level effects are discussed. Those results are pretty interesting, in that different interventions differentially affect the four DVs.

The current paper uses the same data set but now slices them in a somewhat different way, analyzing the effectiveness of the 11 interventions by political ideology of respondents, but only for 3 of the 4 DVs (omitting info sharing willingness). Country-level effects of political polarization on the three DVs (i think for the control groups in each country who received no treatment, is that true?) are shown in the very interesting Figure 2, which is not much discussed in the text.

With a data collection as ambitious and resource intensive as this one, it is completely understandable that the authors will want to try to harvest multiple publications from the effort. It would help reviewers and editors, however, to get a sense of how many papers are planned (in addition to the two we already know about) and in what ways they will differ. Also, how was authorship determined on the two existing (and any future) papers? Why 300+ on paper 1 and only 5 on this one? All of this could be mentioned perhaps in a paragraph or footnote towards the beginning of this paper and in Paper 1. I personally think there is (just about) enough novelty in this paper to warrant separate publication in Nature Communications, but others may disagree, and the reader ought to be informed a bit better on what is new here and where to go for other information on the data.

I don't have many comments or concerns about the paper. It is well written, concise, and presents very interesting results.

There are many very nice features, including the open access of the analysis scripts, the crowd sourcing of the interventions (maybe say a bit more on how that worked and who provided input?), and the use of the consequential, costly WEPT task. There are a few things though that could be improved and clarified, listed below.

Given the centrality of the political polarization construct, it would be good to have some indication of its distribution in the 60 countries. Median split is one way to "standardize", but there are other ways to split the sample in each country, along more theoretical justifications. Maybe do it differently and do a sensitivity analysis?

Table 1: i am surprised that no positive emotions intervention made it into the tested set, given that enhancing pride or hope has been shown in multiple studies to motivate pro-environmental behavior. Can you comment on this omission? Was it not nominated?

Line 146-149: Is the DV here the index of all 4 climate change beliefs or only the "climate change is a global emergency" item. I think it is the former, but the sentence suggests the latter.

Line 240: an extra "e" typo, to be deleted.

Following sentence: The fact that it is Conservatives who plant trees despite their more negative beliefs/attitudes goes against the story suggested earlier that "personal costs" get in the way of attitudes translating into actions.

What exactly is shown in Figure 4, say in the first panel. From this figure it appears that for

conservatives ALL 11 interventions decreased Belief, i.e., the change is negative?!? But that is not what the text says in lines 269ff? and the figure legend should say "change" rather than "increase"....

Line 305: there is a "3" missing before "interventions", i think?

Line 374ff: Any idea why these interventions backfire for personal action?

Line 479f: This is the first time the authors mention that political orientation is measured by two items, i think. Say here or earlier why they were combined?

REVIEWER COMMENTS

Reviewer #1 (Remarks to the Author):

1.1: I found myself a little conflicted reading this paper. On the one hand: what a great dataset, a tribute to the authors' leadership and organizational skills! Also, I liked the big-picture framing, and the paper is presented professionally and clearly (minor asides: a critical number is missing in par. 2 of section 2.2 and in my copy I can't see the promised vertical dotted lines in Fig 4.)

We thank the reviewer for their time and thoughtful feedback! We also appreciate the reviewer's positive assessment of our dataset, big picture framing, and clear overall presentation.

We have now updated paragraph 2 in section 2.2 to include all relevant numbers for interventions that impacted liberals, conservatives, and both, and also updated Figure 4 to include the vertical lines indicating mean values from the control condition.

Fig 4. Interventions' impacts on climate beliefs (Panel A), policy support (Panel B), and action (Panel C), split by political ideology (liberals in blue and conservatives in red). Vertical lines indicate the average in the control condition, for each ideological grouping. Upward triangles indicate significant increases, downward triangles indicate significant decreases, and circles indicate no significant differences, always compared to the ideologically congruent controls.

1.2: On the downside, I'm not sure I feel confident in the two headline conclusions: (1) that differences in political ideology disappear when it comes to costly behavior and (2) that climate change interventions backfire for conservatives. Let me take each of these separately. The paper shows that differences in political ideology – robust when examining climate beliefs and policy support - disappear when it comes to climate action. This is an attention-grabbing conclusion, and is probably the most memorable part of the paper. The takehome for many readers will be that liberals are basically virtue signalling around climate change. People paying closer attention will see that this isn't quite the message: that for whatever reasons conservatives are acting more green than their belief systems would

imply. My main hesitation here is around the operationalisation of climate action as tree-planting. Tree-planting is well-understood to be conservatives' favorite climate action because it doesn't ask people to accept structural changes in how society and business is structured. What conservatives rail against is government regulation that constrains the freedoms of individuals and markets. Climate action that disrupts the status quo, implies lifestyle changes, and challenges the hegemony of fossil fuel vested interests would probably see robust ideological divides. Individual, banded solution that do not disrupt the status quo and have broad green benefits beyond climate change (e.g., greening public spaces) would see smaller ideological divides. A revised paper should carry that caveat and perhaps soften the conclusion accordingly.

We are grateful to the reviewer for this important point, with which we definitely agree. The tree planting task captures an individual-level pro-environmental behavior, which one could argue conservatives might particularly resonate with, especially as an alternative to system-level change. However, the empirical literature on this interpretation is not entirely consistent. For example, recent work has found a negative correlation between political conservatism and tree planting (Essl, Hauser, Suter, & von Bieberstein, 2023). Nevertheless, we have made changes throughout the manuscript to point out the individual nature of the tree planting task as a climate action. We also added the following paragraph to the general discussion section to acknowledge these critical points which are helpful to the interpretation of our results:

*“When assessing whether the conceptual-behavioral disconnect observed was triggered by liberals not acting on their beliefs (e.g., a liberal-oriented “green gap”) or conservatives acting despite their beliefs, we found that conservatives’ beliefs predicted their own behaviors less than liberals’ beliefs predicted liberals’ behaviors. This suggests that the disconnect between beliefs and behaviors in these results could be more strongly driven by conservatives acting in a more pro-environmental manner than their beliefs would predict. This result aligns with recent findings in the United States²³, and is more consistent with a process by which conservatives contributed to tree planting efforts at similar rates as liberals despite not believing in the urgency of climate change as much as liberals. **Instead, conservatives may have acted in this pro-climate way for reasons other than climate concern, such as to preserve and protect traditional values of nature and purity⁶². Alternatively, conservatives may have conceptualized tree planting as an alternative to system-level action on climate. Future research should disentangle these mechanisms, which could help assess the generalizability of these findings to other behaviors.**”*

1.3: The second conclusion is that climate change interventions are largely ineffective with regard to behavior and potentially backfire, especially for conservatives (i.e., psychological interventions mostly don't work and potentially do harm). Here, I worry that the design carries a confound that could help explain the pattern (I should emphasise this criticism applies only to the behavior measure). WEPT requires participants to put in extra effort and time over and above the core business of reading the materials carried in the stimuli and responding to the DVs. The more time-consuming or cognitively taxing the intervention, one might expect that participants will have fewer resources to spare on the WEPT task. In other words, the design may be biased toward showing greater action in conditions that are easy and quick. It's difficult to know how big a problem this is on the basis of the information provided. The control condition required people to read an extract

from Dickens. If the intervention tasks were more onerous than this, it might explain the high level of backfiring observed. I wonder if the authors have access to time data that can help referee whether participants spent as much time on this task as they did on the 11 interventions? Grazing the description of the interventions, some of them almost certainly required more time / investment than others. Psychological distance, future-self continuity, and letter to future generation interventions required participants to do a thinking and writing task, which is much more involved in terms of time and effort than simply reading materials. Eyeballing Fig 4 reinforces my suspicion that there may be a confound here. For example, psychological distance is the MOST effective intervention on beliefs, effective for both liberals and conservatives, but the LEAST effective on behavior, apparently backfiring for both liberals and conservatives. Similarly, the letter to future generations task is the MOST effective intervention on policy support, but the FOURTH LEAST effective on behavior. Either there is something incredibly strange going on here psychologically, or participants were simply exhausted by these interventions and didn't have time for the WEPT.

We thank the reviewer for this insightful observation, and have now conducted additional analyses to shed light on the interpretation of the negative effects observed. To address this possibility that time spent on the interventions impacted WEPT performance, we extracted the time spent in the intervention phase of the experiment by each participant. We used this measure to assess whether the amount of time spent on the interventions impacted the number of trees planted, and whether adjusting for intervention time changes the results.

Descriptively, in terms of time spent, the control condition ranked number 8 out of the 12 possible conditions, meaning that, on average, the control condition took more time than 7 of the interventions, and less time than 4 interventions.

To address the potential impact of time spent on the intervention phase on the number of trees planted we ran a series of additional models in which we included intervention time as a fixed effect, as follows:

1. First, we ran a ordinal mixed model, with trees planted in the WEPT as the DV, condition as it interacts with ideology as a fixed effect and intervention time as another fixed effect, and by-country random intercepts. This model corresponds to Results section 2.3; Table S9, but has the addition of intervention time as another fixed effect, to statistically control for time spent on the interventions when observing the effects of condition and ideology on tree planting. The results from this model can be seen below and have now been included in the supplement (Table S10).

We found a *positive* main effect of intervention time ($\beta = .0008$, $SE = 0.00005$, $z = 17.71$, $p < .001$), suggesting that participants who spent more time in the intervention phase planted more trees. The effects of condition when controlling for time remained mostly unchanged. Most centrally, the effect of ideology on climate action remained statistically non-significant ($p=0.169$). The main differences introduced by controlling for intervention time are for the main effects of the Binding Moral Foundations and Scientific Consensus interventions, both of which have now significantly increased the number of trees planted compared to control.

Generally, these results suggest that our findings are not largely impacted when controlling for intervention time.

Table S10.

Results from ordinal mixed model with trees planted in the WEPT as outcome, the interaction of condition and ideology as the fixed effect, condition time as a fixed effect, with by-country random intercepts.

Fixed Effect	Estimate	SE	z	p
Condition Time	0.0008	0.00005	17.71	< .001
IdeologyLiberal	-0.08	0.06	-1.38	0.169
BindingMoral	0.19	0.06	3.16	0.002
SciConsens	0.14	0.06	2.41	0.016
PluralIgnorance	0.05	0.06	0.89	0.375
DynamicNorm	0.05	0.06	0.78	0.436
SystemJust	-0.02	0.06	-0.34	0.735
CollectAction	-0.13	0.06	-2.25	0.025
FutureSelfCont	-0.30	0.06	-4.87	< .001
WorkTogetherNorm	-0.32	0.06	-5.52	< .001
NegativeEmotions	-0.38	0.06	-6.38	< .001
PsychDistance	-0.45	0.06	-7.58	< .001
LetterFutureGen	-0.50	0.06	-8.01	< .001
DynamicNorm:IdeologyLiberal	0.25	0.08	3.02	0.003
WorkTogetherNorm:IdeologyLiberal	0.24	0.08	2.98	0.003
FutureSelfCont:IdeologyLiberal	0.23	0.09	2.64	0.008
SciConsens:IdeologyLiberal	0.20	0.08	2.49	0.013
SystemJust:IdeologyLiberal	0.19	0.08	2.29	0.022
PsychDistance:IdeologyLiberal	0.17	0.08	2.08	0.038
BindingMoral:IdeologyLiberal	0.16	0.08	1.93	0.053
LetterFutureGen:IdeologyLiberal	0.16	0.09	1.81	0.071
NegativeEmotions:IdeologyLiberal	0.14	0.08	1.77	0.077
CollectAction:IdeologyLiberal	0.12	0.08	1.48	0.140
PluralIgnorance:IdeologyLiberal	0.12	0.08	1.44	0.149

- Next, we also assessed the interaction between intervention time and the interventions' effects. To investigate, we ran another ordinal mixed model predicting the number of

trees planted but now as a function of condition as it interacts with intervention time, including country random intercepts (Table S11, reported below). We again found that on average intervention time positively predicts the number of trees planted. When assessing the interactions, we found that for 2 of the interventions (binding moral foundations and scientific consensus), the more time participants spent the more trees they planted compared to the control. However, we found that for 8 of the interventions, the more time participants spent in the intervention phase the *fewer* trees they planted compared to control.

These results provide support for the reviewer's intuition, that after these 8 interventions, participants might have planted fewer trees because they could have been exhausted, given the time budget spent on the intervention phase. We have now added these analyses and interpretations to the manuscript.

Table S11.

Results from ordinal mixed model with trees planted in the WEPT as outcome, the interaction of intervention time and condition as the fixed effect, with by-country random intercepts.

Fixed Effect	Estimate	SE	z	p
Condition Time	0.002	0.000	10.921	< .001
SciConsens	0.449	0.063	7.108	< .001
DynamicNorm	0.428	0.064	6.727	< .001
PluralIgnorance	0.411	0.062	6.636	< .001
SystemJust	0.371	0.064	5.821	< .001
CollectAction	0.196	0.066	2.977	0.003
WorkTogetherNorm	0.189	0.066	2.858	0.004
FutureSelfCont	0.165	0.068	2.414	0.016
NegativeEmotions	0.060	0.067	0.897	0.370
LetterFutureGen	-0.021	0.073	-0.285	0.775
PsychDistance	-0.106	0.071	-1.495	0.135
BindingMoral	-0.688	0.062	-11.052	< .001
BindingMoral:Condition Time	0.804	0.021	37.410	< .001

SciConsens:Condition Time	0.003	0.001	2.748	0.006
DynamicNorm:Condition Time	0.000	0.000	-1.202	0.229
PluralIgnorance:Condition Time	-0.001	0.000	-2.539	0.011
SystemJust:Condition Time	-0.001	0.000	-3.914	< .001
CollectAction:Condition Time	-0.001	0.000	-4.403	< .001
PsychDistance:Condition Time	-0.001	0.000	-4.910	< .001
NegativeEmotions:Condition Time	-0.001	0.000	-6.385	< .001
LetterFutureGen:Condition Time	-0.001	0.000	-6.472	< .001
FutureSelfCont:Condition Time	-0.001	0.000	-6.501	< .001
WorkTogetherNorm:Condition Time	-0.002	0.000	-6.571	< .001

Overall, in these models, we find that, on average, spending more time on the intervention phase predicts more trees being planted in the WEPT. However, this average effect at the study level manifested differently within the experimental conditions. Specifically, spending more time on the intervention phase significantly *increased* the number of trees planted in the scientific consensus and binding moral foundations interventions, did *not* affect the tree planting in dynamic norms, and *decreased* the number of trees planted in the other 8 interventions.

This suggests that in the absence of time constraints, some interventions might increase pro-environmental behavior. However, the degree to which these findings actually generalize to pro-environmental behaviors that do not hinge on time (e.g., donations) should still be empirically assessed in future studies.

These results also suggest a complex moderating effect of the length of the interventions and participants' behavior in the tree planting task, unveiling the contexts in which time spent on the intervention played a *positive* versus a *negative* role on the outcome. We believe these insights are critical to the interpretation of our results, and we thank the reviewer for encouraging us to explore this direction more deeply. We have now added this analysis to the Supplemental Information and mentioned it in the results section:

“To further unpack the potential interactive effects of the time spent on the interventions on the number of trees planted, a task that also required time, we also ran an ordinal mixed model with number of trees planted as the dependent variable, condition as it interacts with intervention time as the fixed effects, including by-country random effects (Table S11). We found that, on average, spending more time on the intervention phase predicts more trees being planted in the WEPT. However, this average effect manifested differently within the experimental conditions. Specifically, spending more time on the intervention phase increased the number of

trees planted in the scientific consensus and binding moral foundations interventions, did not affect the tree planting in the dynamic norms intervention, and decreased the number of trees planted in all the other 8 interventions. These results suggest that a reason for the negative effects observed might have been due to the limited time budget participants had for the study, such that less time was allocated to planting trees in the conditions with longer interventions. The results also suggest that in the absence of time constraints, some interventions (e.g., scientific consensus and binding moral foundations) might increase individual level pro-environmental behavior. However, the degree to which these findings actually generalize to behaviors that do not hinge on time (e.g., direct donations) should still be empirically assessed in future studies.”

3. For completeness, we also report the full three-way interaction of condition, ideology, and condition time here, as well as in the supplement (Table S16), although we generally didn't find these interaction effects further interacted with ideology.

Results from ordinal mixed model with trees planted in the WEPT as outcome, the interaction of intervention time, condition, and ideology as the fixed effect, with by-country random intercepts.

Fixed Effect	Estimate	SE	z	p
Condition Time	0.002	0.000	5.88	< .001
IdeologyLiberal	-0.096	0.132	-0.73	0.465
PluralIgnorance	0.413	0.106	3.92	< .001
DynamicNorm	0.369	0.109	3.39	< .001
SystemJust	0.212	0.113	1.87	0.061
CollectAction	0.167	0.115	1.45	0.147
FutureSelfCont	0.140	0.116	1.20	0.230
WorkTogetherNorm	0.105	0.112	0.94	0.349
SciConsens	0.110	0.119	0.92	0.357
NegativeEmotions	0.073	0.114	0.64	0.524
LetterFutureGen	-0.125	0.124	-1.01	0.313
PsychDistance	-0.261	0.122	-2.14	0.033

Fixed Effect	Estimate	SE	z	p
BindingMoral	-0.758	0.109	-6.94	< .001
BindingMoral:Condition Time	0.833	0.034	24.46	< .001
SciConsens:Condition Time	0.018	0.003	5.37	< .001
SystemJust:Condition Time	-0.001	0.001	-1.17	0.243
PsychDistance:Condition Time	-0.001	0.000	-2.45	0.014
DynamicNorm:Condition Time	-0.001	0.001	-2.53	0.011
CollectAction:Condition Time	-0.001	0.000	-2.90	0.004
LetterFutureGen:Condition Time	-0.002	0.000	-3.73	< .001
FutureSelfCont:Condition Time	-0.002	0.000	-4.28	< .001
PluralIgnorance:Condition Time	-0.002	0.000	-4.39	< .001
WorkTogetherNorm:Condition Time	-0.002	0.000	-4.44	< .001
NegativeEmotions:Condition Time	-0.002	0.000	-4.50	< .001
SciConsens:IdeologyLiberal	0.573	0.156	3.68	< .001
PsychDistance:IdeologyLiberal	0.369	0.168	2.20	0.028
SystemJust:IdeologyLiberal	0.317	0.154	2.05	0.040
CollectAction:IdeologyLiberal	0.213	0.157	1.36	0.175
LetterFutureGen:IdeologyLiberal	0.221	0.171	1.30	0.195
DynamicNorm:IdeologyLiberal	0.174	0.153	1.14	0.254
FutureSelfCont:IdeologyLiberal	0.138	0.163	0.84	0.400
BindingMoral:IdeologyLiberal	0.123	0.150	0.82	0.414
NegativeEmotions:IdeologyLiberal	0.066	0.158	0.42	0.677
PluralIgnorance:IdeologyLiberal	-0.031	0.150	-0.21	0.837
Condition Time:IdeologyLiberal	0.000	0.001	-0.22	0.824

Fixed Effect	Estimate	SE	z	p
WorkTogetherNorm:IdeologyLiberal	-0.040	0.162	-0.25	0.807
PluralIgnorance:Condition Time:IdeologyLiberal	0.003	0.001	3.16	0.002
WorkTogetherNorm:Condition Time:IdeologyLiberal	0.002	0.001	2.47	0.013
DynamicNorm:Condition Time:IdeologyLiberal	0.001	0.001	1.53	0.126
BindingMoral:Condition Time:IdeologyLiberal	0.039	0.049	0.80	0.421
NegativeEmotions:Condition Time:IdeologyLiberal	0.000	0.001	0.79	0.430
FutureSelfCont:Condition Time:IdeologyLiberal	0.000	0.001	0.74	0.458
LetterFutureGen:Condition Time:IdeologyLiberal	0.000	0.001	0.02	0.987
CollectAction:Condition Time:IdeologyLiberal	0.000	0.001	-0.42	0.672
PsychDistance:Condition Time:IdeologyLiberal	0.000	0.001	-0.73	0.468
SystemJust:Condition Time:IdeologyLiberal	-0.001	0.001	-1.22	0.224
SciConsens:Condition Time:IdeologyLiberal	-0.019	0.003	-5.44	< .001

1.4: In conclusion, I'm definitely intrigued by the data and open-minded to both takehome messages. But these messages are combustive ones and so the authors would want to make sure they are robust and that the paper carries all the necessary nuance.

We appreciate this positive feedback and have taken efforts to more fully and effectively communicate the nuance in our results in the discussion. We welcome any further suggestions for adding nuance to the paper.

1.5: Three final points of clarification. First, it looks like this is a secondary analysis separate from another manuscript currently under review but drawing on the same dataset. It would be useful to know what that other paper was about so reviewers and editors can judge the scale of conceptual overlap.

We agree, and have included the entirety of the other manuscript being prepared for publication in the submission files with the manuscript so reviewers and editors can assess the similarities and differences. In short, the other paper investigates how effective/ineffective the interventions are across the dependent variables, and along estimations of initial levels of belief, but does not investigate political ideology in any way. As such, we think the contributions of these two papers are distinct and will largely be interesting to very different audiences.

Moreover, it would likely be an unwieldy paper to write if we included all the results in a single paper (readers would likely lose the plot and struggle to see the main contribution). As such, we thought it was much better for readers to see them separately as they make independent contributions to different literatures (e.g., political scientists, sociologists and other social scientists who study polarization and ideology will uniquely be interested in the current paper, but not the other paper).

1.6: Second, it looks like there was massive attrition (>80k respondents whittle down to <55k usable participants). Any thoughts on why, and what this means for the integrity of the dataset?

We thank the reviewer for pointing out this issue. The difference between the number of participants who took the study and the number who passed the pre-registered attention checks is large (~30K participants), the rate (36.4%) is fairly normal for online surveys. Research suggests that online studies do suffer from higher attrition rates compared to laboratory studies, typically ranging from around 30% — 50% (for review, see Aguinis, Villamor, & Ramani, 2021). Beyond the attention check failures, we also excluded participants who did not provide an answer on the ideological measure (as well as in countries where the ethics boards would not allow us to include this measure), which was the focus of this study (around 4K participants). We now added these details to the methods section:

“Participants’ data came from a previously collected dataset⁴⁵. A total of 83,927 completed the study. Of them, 59,440 participants ($M_{age}=39.13$, $SD_{age}=15.76$; 50% women, 46% men) from 63 countries passed the two attention checks (i.e., Please select the color “purple” from the list below.” and “To indicate you are reading this paragraph, please type the word sixty in the text box below.”) and correctly completed the WEPT demo. Of them, ideological information was therefore available for 51,224 participants ($M_{age} = 39.62$, $SD_{age} = 15.82$; 50% women, 49% men, 1% non-binary), from 60 countries. The control condition contained a sample of 4,302 participants ($M_{age} = 39.52$, $SD_{age} = 15.64$; 52% women, 47% men, 1% non-binary).”

We also mentioned attrition as a limitation in the discussion section:

“Another limitation of the current study is the attrition rate observed (36.4%) between the number of participants who completed the study and the participants whose data ended up in the final analysis. Although within the expected range of attrition for online studies (30% — 50%; Aguinis, Villamor, & Ramani, 2021), this feature of our data should be considered when interpreting and generalizing the results. Future research should thus replicate the current findings in more controlled environments that may benefit from lower attrition rates.”

1.7: Third, what does it mean to say that 69% of countries included representative samples on at least one demographic variable? It might be better to just accept that these are convenience samples rather than straining too hard to make a case for representativeness.

We appreciate the feedback and have incorporated this suggestion in the Methods and Introduction.

Reviewer #2 (Remarks to the Author):

2.1: This paper reports the results of a mega-study on climate change interventions. It also makes very good use of the large control group to arrive at some important insights about the correlates of beliefs, policy support, and behavior. There are several crucial lessons, perhaps most importantly about the need to carefully differentiate outcome variables – both when looking at correlates and considering interventions. I was aware of this project, and it was a pleasure to read the paper. I support publication and have just a few suggestions for the authors to consider.

We are very grateful to the reviewer for their time and insightful suggestions. We also greatly appreciate that you support publication!

2.2: First, there is some other literature that has shown differences across outcome variables that may be worth considering. This is summarized more less in: https://www.tandfonline.com/doi/full/10.1080/17524032.2020.1805343?casa_token=gL28KT P96aIAAAAA%3AuMIPGucbAFFI9sOBrc6Yl8En2R0cHrmS-mKkSjbsSO2C3IsDI3MqInpYM4Opjf223kCzeFcW9Z0GCNY. Put another way, prior work also had shown distinct effects on outcomes (also see, e.g., Key and Campbell 2014). This study is a better demonstration, but it may be worth connecting it to that prior discussion.

We thank the reviewer for this suggestion. We have now incorporated this literature in our general discussion section:

“We also found ideological differences in the impact of climate change interventions on liberals’ and conservatives’ climate beliefs, policy support, and action. Critically, these ideological effects differed across these outcomes, consistent with prior work pointing to the critical importance of the outcome when assessing the effectiveness of climate interventions (Bayes et al., 2020).”

2.3: Second, the other literature which is worth mentioning is the debate on backfiring effects between van der Linden and Kahan—the findings here raise interesting questions about the psychological processes at work.

We appreciate this suggestion. Kahan (2017) argues that scientific consensus messaging renders null effects on beliefs and policy support, which he finds when reanalyzing the data posted by van der Linden and colleagues (2015). While our dataset could speak to this important debate, the current paper is more concerned with ideological differences in the responses to the interventions, rather than with the presence or absence of main effects. We are hesitant to include main effects results interpretations here, to avoid overlap with the main collaboration paper (see our response to the previous reviewer). Instead, we have referenced the other paper more explicitly and encourage interested readers to check it out. Nevertheless, we had originally

included the debate on whether scientific consensus backfires for climate change skeptics in our introduction as the justification for our current investigation:

“Indeed, climate skeptics—compared to climate believers—have been found to respond differently to climate interventions^{37,38}. For example, scientific consensus messaging (i.e., informing the public that most scientists are in agreement about the climate crisis^{39,40}), has limited effects on climate skeptics’ support for climate action^{41,42,43}, or can even spark reactance and decrease support for climate policy^{37,44}.”

We have now added the following paragraph to the discussion section, to connect this debate to our findings:

“However, when it comes to stimulating individual-level climate action, our findings suggest that practitioners could successfully deploy scientific consensus messaging, but only when targeting liberals. This finding aligns with prior work suggesting that scientific consensus messaging has limited effects on climate skeptics’ support for climate action^{41,42,43}. “

2.4: Third, I would be interested in whether the “taking action...” belief variable demonstrated any distinct dynamics from the other three items in that composite given it involves action and is in fact fairly similar to what the gateway belief model uses for climate action. This may be worth a footnote (given the tree planting results).

In response to this insightful question, we conducted a factor analysis on our four-item measure of belief in climate change, which included the item "Taking action to fight climate change is necessary to avoid a global catastrophe." The purpose was to discern if the “Taking action...” item exhibited unique dynamics compared to the other three items, especially since it pertains to proactive measures and shares similarities with the Gateway Belief Model's emphasis on climate action. To assess the suitability of our data for factor analysis, we first calculated the Kaiser-Meyer-Olkin (KMO) measure of sampling adequacy. Our obtained KMO value was 0.862, which is considered 'great'. This suggests that our dataset was indeed appropriate for a factor analysis. The results from our factor analysis, as supported by the scree plot and eigenvalue criterion, indicated that all four belief variables loaded highly onto a single factor, suggesting they collectively measure a cohesive underlying construct. This implies that the “Taking action...” item did not demonstrate distinct dynamics separate from the other belief items. Given the strong factor loadings, we can conclude that these four items can be safely used as a composite score for measuring belief in climate change. Finally, the overall Cronbach's for this scale ($\alpha = 0.87$) suggests that this scale is highly reliable.

2.5: Fourth, did the authors check an interaction with age (and ideology) in the correlational results as some prior work suggests younger conservatives are more supportive of climate policies and actions (e.g., see Pew polling).

We thank the reviewer for this insightful suggestion, and we share the interest in this question. In line with the Pew polling results the reviewer is referring to, we also found a significant ideology by age interaction in predicting beliefs and policy support. That is, the older a participant was, the more their conservative ideology decreased their beliefs in climate change ($\beta = -.006$, $SE = 0.001$, $t = -6.44$, $p < .001$, $d = 0.21$) and policy support ($\beta = -.005$, $SE = 0.001$, $t = -5.923$, $p < .001$, $d = 0.19$). This interaction was not significant in predicting tree planting ($p = .414$). We now report these findings in the supplement, and refer to them in the main results section:

“The effect of ideology on both beliefs and policy support interacted with age, such that the conservative ideology was associated with decreased belief and policy support more so for older participants (Table S13).”

2.6: Fifth, I found the discussion around Figure 3 a bit too sparse.

We appreciate the need for more discussion around this set of results. As such, the full discussion around this point has been expanded in the general discussion section:

“When assessing whether the conceptual-behavioral disconnect observed was triggered by liberals not acting on their beliefs (e.g., a liberal-oriented “green gap”) or conservatives acting despite their beliefs, we found that conservatives’ beliefs predicted their own behaviors less than liberals’ beliefs predicted liberals’ behaviors. This suggests that the disconnect between beliefs and behaviors in these results could be more strongly driven by conservatives acting in a more pro-environmental manner than their beliefs would predict. This result aligns with recent findings in the United States²³, and is more consistent with a process by which

conservatives contributed to tree planting efforts at similar rates as liberals despite not believing in the urgency of climate change as much as liberals. Instead, conservatives may have acted in this pro-climate way for reasons other than climate concern, such as to preserve and protect traditional values of nature and purity⁶². Alternatively, conservatives may have conceptualized tree planting as an alternative to system-level action on climate. Future research should disentangle these mechanisms, which could help assess the generalizability of these findings to other behaviors.”

2.7: Sixth, I really liked the exploration of the relationship between beliefs and behaviors in terms of whether liberals are not acting or conservative acting despite beliefs. Can the authors do the same thing with policy—as a precursor to behavior and a consequence of belief (particularly the former)? Even if there is no relationship in that case, it would be interesting to me.

We appreciate the reviewer’s interest in this analysis. We ran an ordinal mixed model with number of trees planted in the behavioral task as the dependent variable, including political ideology (continuously measured from 0 = liberal to 100 = conservative) as it interacts with climate policy support as the fixed effect, and by-country random effects. We found a significant main effect of policy support ($\beta = 0.022$, $SE = 0.003$, $z = 6.18$, $p < .001$, $d = 0.17$) as well as a significant ideology by policy support interaction ($\beta = -0.0001$, $SE = 0.00005$, $z = -2.97$, $p = .003$), such that the more conservative the participant, the less their climate policy support predicted their climate behaviors. These results are virtually the same as those we found for belief. We have added a line addressing this to our manuscript (Results section 1.4):

“It is not just beliefs that better predict behavior for liberals compared to conservatives – policy support is also more strongly associated with behavior for liberals compared to conservatives (i.e., suggested by an ideology by policy interaction when predicting behavior: $\beta = -0.0002$, $SE = 0.00006$, $z = -2.966$, $p = 0.003$).”

2.8: Seventh, my understanding is that none of the interventions allowed for targeted messages for liberals and conservatives (all were homogenous treatments)? If that is correct, I would make it clearer. Along those lines, my most salient point is that I think one implication of the results is that climate communicators need to consider generalized versus targeted messages across three distinct outcomes. That results in a 2x3 framework that probably should be explored in future work and scholars need to stop bundling messages and outcomes without being explicit.

We thank the reviewer for raising this important point. Indeed, all treatments tested in our study were homogeneously administered without targeting. We agree this should be made clearer in the paper, so we have revised the manuscript accordingly. Specifically, we added the following paragraph to the discussion section:

“Notably, the interventions tested here were randomly assigned to participants, and not ideologically targeted. Given recent work showing that targeted interventions can be up to 200%

more effective (Tappin et al., 2023) we hope our results can inform and increase the potential of future targeted interventions on climate change. Moreover, given the heterogeneity of the effects observed across ideologies and outcomes, we also recommend future research aimed at targeting mechanisms to consider these dimensions for optimal impact.”

2.9: Finally, a small point – given pre-registration and in essence this is 11 distinct experiments, I do not think multiple comparisons are a problem, but the authors may want to mention it in the methods section to address the point as others may think of it.

We appreciate this suggestion. While the main effect analyses were pre-registered, the ideological interaction analyses were not. However, the degrees of freedom in models analyzing the 11 conditions together are distinct from the degrees of freedom obtained when running 11 independent experiments. In each analysis reported here, our models partially pool variance from the other conditions when looking at the effect of a given condition compared to control. This feature of the mixed (multilevel) models employed here guards against the multiple comparison issue which would have been a problem had we conducted independent analyses between each intervention and the control. This explanation can be found in a classic chapter by Gelman and colleagues (2009). We have now added the following paragraph to the methods section clarifying this important point:

“4 Statistical Methods. Throughout the results section we used hierarchical mixed effects models to assess the effects of interest. Of note, these multilevel models used to account for the data being non-independent within countries, participants and items (for beliefs and policy support) have the benefit of alleviating multiple comparison concerns by performing partial pooling in generating estimates (Gelman, Hill, & Yajima, 2012).”

Reviewer #3 (Remarks to the Author):

3.1: This appears to be the second paper to be published based on a herculean data collection effort in 60 countries. The first one (Reference 45 in the current manuscript and also attached for the reviewers' benefit, with an author count of about 300) reports on the effectiveness of 11 interventions to increase climate concern and action, crowdsourced by experts and tested in a between-subject design, on four DVs: belief in climate change, support for climate policy, willingness to share climate information on the internet, and a costly personal behavior. These results are further unpacked for respondents who differ in their initial belief in climate change, and country level effects are discussed. Those results are pretty interesting, in that different interventions differentially affect the four DVs. The current paper uses the same data set but now slices them in a somewhat different way, analyzing the effectiveness of the 11 interventions by political ideology of respondents, but only for 3 of the 4 DVs (omitting info sharing willingness). Country-level effects of political polarization on the three DVs (i think for the control groups in each country who received

no treatment, is that true?) are shown in the very interesting Figure 2, which is not much discussed in the text.

We thank the reviewer for their time and insightful suggestions. We are glad they found our efforts worthwhile.

Regarding Fig. 2, the reviewer is correct to note this set of figures corresponds to the control condition, in which participants did not receive a climate change intervention (instead they read an excerpt of unrelated text). We updated the figure legend to make this more clear. This figure is intended to orient readers to the scale of polarization globally in a descriptive way, and is accompanied by the point plots in the supplement (Fig S1-S6).

Indeed, we do not discuss these country-level polarization differences in depth in this manuscript beyond describing them to generally orient the readers, for two reasons. First, the focus of our paper is on aggregate levels of ideological differences on climate outcomes and intervention effects. Delving into country-specific effects would exponentially increase the complexity of this investigation, which we believe might distract from our main points. Second, a different subset of our co-authors on the original collaboration have pre-registered an analysis of the country-specific predictors of political polarization, so we are also hoping to avoid overlap with their study.

3.2: With a data collection as ambitious and resource intensive as this one, it is completely understandable that the authors will want to try to harvest multiple publications from the effort. It would help reviewers and editors, however, to get a sense of how many papers are planned (in addition to the two we already know about) and in what ways they will differ. Also, how was authorship determined on the two existing (and any future) papers? Why 300+ on paper 1 and only 5 on this one? All of this could be mentioned perhaps in a paragraph or footnote towards the beginning of this paper and in Paper 1. I personally think there is (just about) enough novelty in this paper to warrant separate publication in Nature Communications, but others may disagree, and the reader ought to be informed a bit better on what is new here and where to go for other information on the data.

We appreciate and agree with the need for transparency and consistency in publication practices. The main paper includes all 255 coauthors. All of them were then prompted to sign up for conducting followup analyses on the dataset given their interests. For example, we signed up to conduct ideological investigations, others signed up to look into other variables (e.g., age, gender, cultural differences etc.). Here is the full list of pre-registered follow-up investigations by our coauthors so far:

https://docs.google.com/document/d/1LkIGTVcUD6b16n9j6OM6ZDm93krteFf_geXubT_fHws/edit

Specifically, for the main paper, we had strict guidelines for conferring authorship, which we established before launching the call for collaboration in 2021, as detailed on our project's website: <https://manylabsclimate.wordpress.com/call-for-collaboration/>

We confirmed each coauthor contributed according to these pre-established guidelines. Specifically, we limited authorship to 2 co-authors per lab/team. Each lab/team could join the project in one of 3 ways:

- First, by collecting data in a new country, including translating the entire study, adapting the stimuli to the local context, obtaining ethics approval, collecting and cleaning the data.
- The second path to co-authorship was by assisting a team of researchers willing and able to do the first path, but lacking financial resources. These teams were paired together and instructed to collaborate on every stage.
- The third and final way to contribute was to have proposed one of the 11 interventions which we included in the final study. This involved suggesting the intervention, then building it in concrete terms (e.g., creating stimuli, programming it, working with the leadership team to adapt it to each cultural context in which we collected data).
- All co-authors also helped edit the main manuscript drafts. All of these co-authors were listed in alphabetical order.
- The leadership team is listed at the beginning of the author block, in order of contribution. The leadership team conceptualized and coordinated the main project, at every stage.

The main paper now has 255 coauthors, who will also be co-authors on a data paper describing the dataset. These co-authors have also been offered consortia co-authorship on a third publication in which we forecasted the main results (which has not yet been submitted for publication).

Beyond these three main papers, our collaborators were promised early access to the dataset to test their own secondary hypotheses. These secondary projects carry no assumption of authorship beyond the specific research team involved in drafting a pre-registration and publishing the research.

Beyond our collaborators, this dataset is intended to become a widely accessible public resource for the behavioral science community at large, to investigate even more questions surrounding climate change interventions across the world. We hope it inspires dozens of researchers to conduct unique analyses to share with the scientific community. We see the dataset as a major public good, in this respect.

3.3: I don't have many comments or concerns about the paper. It is well written, concise, and presents very interesting results. There are many very nice features, including the open access of the analysis scripts, the crowd sourcing of the interventions (maybe say a bit more on how that worked and who provided input?), and the use of the consequential, costly WEPT task. There are a few things though that could be improved and clarified, listed below.

We thank the reviewer for noting our work is “well-written, concise, and presents very interesting results”.

To clarify the crowdsourcing process, we added to the following paragraphs to describe the procedure (but note that the detailed explanation of the entire collaboration and crowdsourcing procedure is described in the main paper that reported this collaboration - ref 45):

“To assess whether engaging in climate action is ideologically polarized, and whether partisans respond differently to climate interventions, we analyzed the ideological differences of a dataset collected as part of an international collaboration that empirically tested the efficacy of eleven climate action interventions (Table 1) compared to a no-intervention control condition, at increasing climate change beliefs and behaviors, in a large and diverse sample⁴⁵. The interventions tested were crowdsourced from behavioral scientists from around the world who answered a call for collaboration posted on professional societies listservs, forums, and on social media. The submitted interventions were screened for feasibility in an international context, relevance for the dependent variables, theoretical support from prior work, and other study specific constraints (e.g., not involving deception, being administrable within 5 minutes, etc). Then, a sample of 188 behavioral science experts rated them on theoretical relevance and practical potential to climate mitigation. To implement the top interventions identified using this process, we then closely worked with the researchers who proposed each intervention, and researchers who had published theoretical work relevant to each intervention, while seeking input from all 250 coauthors on the original study⁴⁵.

Importantly, several interventions are directly relevant to the issue of political polarization given prior work suggesting they might differentially affect liberals and conservatives (e.g., scientific consensus⁴¹), and some were specifically developed based on theoretical work in political psychology (e.g., system justification⁴⁶, or binding moral foundations³⁸).”

3.4: Given the centrality of the political polarization construct, it would be good to have some indication of its distribution in the 60 countries. Median split is one way to "standardize", but there are other ways to split the sample in each country, along more theoretical justifications. Maybe do it differently and do a sensitivity analysis?

We thank the reviewer for this suggestion. Figure 2 and Figures S1-S6 were included to provide a description of the degree of polarization in each country, for each outcome.

Regarding the sensitivity analysis, we agree there are many ways one could split the ideology variable. In addition to the median split within-country, we now added another type of split by ideology informed by the psychometric justification of considering the halfway point of the ideology scale administered. Specifically, the scales participants interacted with were 100-pt sliders ranging from liberal to conservative. Considering participants from 1-50 as liberal and 51-100 as conservative would offer a different approach to the operationalization of political ideology. We considered this option, but found it to be less than ideal because some countries did not use the full scale (e.g., all participants ranging from 60 to 100). As such, we believe this approach to be a less sensitive measure, but have now replicated the results using this definition of ideology. The results from this additional analysis are presented here. As can be seen below (and included in the supplemental materials, Figure S8), results largely remained unchanged. As such, we have great confidence in our original inferences.

Per-country median split

	Climate Beliefs		Climate Policy Support		Climate Action	
	Liberals	Conservatives	Liberals	Conservatives	Liberals	Conservatives
Psych Distance	✓	✓	✓		✗	✗
Letter to Future Gen.	✓	✓	✓	✓	✗	✗
Collective Action	✓	✓	✓	✓		✗
Future-Self Cont.	✓	✓	✓	✓		✗
Negative Emotions				✗	✗	✗
System Justification	✓	✓	✓			✗
Scientific Consensus					✓	
Binding Moral Found.	✓					
Dynamic Norms						
Pluralistic Ignorance						✗
Work TogetherNorm					✗	✗

Ideology scale midpoint split

	Climate Beliefs		Climate Policy Support		Climate Action	
	Liberals	Conservatives	Liberals	Conservatives	Liberals	Conservatives
Psych Distance	✓	✓	✓		✗	✗
Letter to Future Gen.		✓	✓	✓	✗	✗
Collective Action	✓	✓	✓	✓		✗
Future-Self Cont.	✓	✓	✓	✓		✗
Negative Emotions				✗	✗	✗
System Justification	✓	✓				✗
Scientific Consensus					✓	
Binding Moral Found.						
Dynamic Norms						
Pluralistic Ignorance						✗
Work TogetherNorm					✗	✗

3.5: Table 1: i am surprised that no positive emotions intervention made it into the tested set, given that enhancing pride or hope has been shown in multiple studies to motivate pro-environmental behavior. Can you comment on this omission? Was it not nominated?

We appreciate this inquiry into the interventions. We share the reviewer’s view regarding the importance of positive emotions such as hope. Indeed, the “Effective Collective Action” intervention was specifically designed to evoke feelings of hope and efficacy both at the individual and collective levels (Goldenberg et al., 2018). This intervention featured examples of successful climate actions that individuals and group took in the past, which have had meaningful impacts in addressing the climate crisis. We realize the confusion may arise from the name assigned to this intervention - instead of naming it “positive emotion” to more clearly parallel the “negative emotion” intervention, we decided to name this intervention “effective collective action” in order to stick to the convention in this literature. However, to more clearly emphasize the positive emotion nature of this intervention, we now updated the description of this intervention in Table 1 to help readers understand that we included positive emotion in this intervention, adding:

“Effective Collective Action: Features examples of successful collective action that have had meaningful effects on climate policies (e.g., protests) or have solved past global issues (e.g., the restoration of the ozone layer) with the scope of increasing positive emotions and efficacy.”

3.6: Line 146-149: Is the DV here the index of all 4 climate change beliefs or only the "climate change is a global emergency" item. I think it is the former, but the sentence suggests the latter.

We thank the reviewer for raising this clarification point. It is indeed the former. These analyses include all the belief items. We have edited this sentence to clarify this point:

“We found a significant effect of ideology ($\beta = -0.19$, $SE = 0.02$, $t = -11.47$, $p < .001$, $d = 0.35$), such that the more participants identified with a liberal ideology, the more they believed in climate change than conservatives (Fig 1A).”

3.7: Line 240: an extra "e" typo, to be deleted.

We thank the reviewer for pointing out this error, which we have now corrected.

3.8: Following sentence: The fact that it is Conservatives who plant trees despite their more negative beliefs/attitudes goes against the story suggested earlier that "personal costs" get in the way of attitudes translating into actions.

There is certainly nuance in these findings regarding personal costs to action in this context, for both conservatives and liberals, and we appreciate the chance to offer more clarity here. We find our results consistent with (albeit not entirely explained by) the large body of work showing the moderating effect of personal costs on the relation between attitudes and behavior (Fazio, 1990; Ajzen, & Fishbein, 1980; Glasford, 2022). We posit in the introduction that although personal costs are an important moderating factor in the connection between attitudes and behaviors, it is surely not the only one, and there are several other important factors at play in this dynamic relation.

In our current work, there is evidence for mismatches between attitudes and behaviors for both liberals and conservatives. We now added several paragraphs to the discussion to make these points clearer:

“When assessing whether the conceptual-behavioral disconnect observed was triggered by liberals not acting on their beliefs (e.g., a liberal-oriented “green gap”) or conservatives acting despite their beliefs (e.g., a conservative-oriented “green gap”), we found that conservatives’ beliefs predicted their own behaviors less than liberals’ beliefs predicted liberals’ behaviors. This suggests that the disconnect between beliefs and behaviors in these results could be more strongly driven by conservatives acting in a more pro-environmental manner than their beliefs would predict. This result aligns with recent findings in the United States²³, and is more consistent with a process by which conservatives contributed to tree planting efforts despite not believing in the urgency of climate change as much as liberals. Instead, conservatives may have acted in this pro-climate way for reasons other than climate concern, such as to preserve and protect traditional values of nature and purity⁶³. Alternatively, conservatives may have conceptualized tree planting as an alternative to system-level action on climate. Future research should disentangle these processes, which could help assess the generalizability of these findings to other behaviors, both at the individual as well as at the collective or systemic levels.”

“Given the importance of political polarization in addressing climate change, these findings also advance theorizing. First, we provide additional explanations for the green gap phenomenon, previously discussed as liberals’ failure to act on their pro-environmental beliefs. Here, we find the green gap can also arise through the converse process—conservatives’ individual level climate actions (e.g., planting trees) can be elicited in spite of their climate change skepticism. Additional evidence for eliciting climate action without attempting to change beliefs comes from the interventions’ effects on conservatives’ action. That is, most interventions decreased the number of trees conservatives planted compared to control, suggesting that when framed as climate change solutions, conservatives engage in pro-environmental behaviors to a

lesser extent. Thus, in future work we are interested in exploring alternative processes for eliciting conservatives' pro-environmental behaviors, that don't involve their climate beliefs."

3.9: What exactly is shown in Figure 4, say in the first panel. From this figure it appears that for conservatives ALL 11 interventions decreased Belief, i.e., the change is negative?!? But that is not what the text says in lines 269ff? and the figure legend should say "change" rather than "increase"....

We appreciate the need for greater clarity in this figure. As noted by Reviewer 1 in response 1.1, there was a vertical line missing from the figure in our initial submission that aids in interpretation. Each panel shows both liberals' and conservatives' beliefs (Panel A), policy endorsements (Panel B) and behavior (Panel C). Looking at Panel A, the relative difference between the blue and red dots show the degree of difference between liberals and conservatives for each intervention on climate beliefs. The impacts of each intervention for each political ideology can be seen by comparing one dot to the respective vertical line (not by comparing the red dot to the blue dot, which instead reveals the effect of ideology per condition). We hope this explanation suffices, and have added more information to the figure legend in hopes of clarifying. Further, in line with the suggested change, we have edited the figure caption to refer to these panels as interventions' "impacts on" rather than "increases."

Fig 4. Interventions' impacts on climate beliefs (Panel A), policy support (Panel B), and action (Panel C), split by political ideology (liberals in blue and conservatives in red). Vertical lines indicate the average in the control condition, for each ideological grouping. Upward triangles indicate significant increases, downward triangles indicate significant decreases, and circles indicate no significant differences, always compared to the ideologically congruent controls.

3.10: Line 305: there is a "3" missing before "interventions", i think?

We thank the reviewer for spotting this problem, we have now edited this sentence to include the number of interventions (3).

3.11: Line 374ff: Any idea why these interventions backfire for personal action?

We addressed this question in depts in our response to reviewer 1, by running additional models controlling for intervention time and interacting condition with intervention time.

3.12: Line 479f: This is the first time the authors mention that political orientation is measured by two items, i think. Say here or earlier why they were combined?

We opted for these two items to capture political ideology since this construct is often discussed and understood in terms of both social and fiscal/economic components. However, in the absence of distinct hypotheses specific to each of these components in our current investigation, and given a robust positive correlation between the two items ($r = .71, p < .001$), we used the composite score of these two questions, a prevalent procedure in this literature. We have now added this reasoning to the methods section:

“To create an aggregated ideological leaning measure, we took the average of participants’ answers on the two political orientation questions, and treated that as the continuous measure of ideology (from liberal to conservative). We aggregated these two political ideology measures given a robust positive correlation between the two items ($r = .71, p < .001$), a prevalent procedure in this literature (e.g., Everett, 2013; Kteily, Rocklage, McClanahan, & Ho, 2019).”

Reviewer #2 (Remarks to the Author):

I appreciate the authors' attention to my comments and those of the other reviewers. I also do believe this paper is sufficiently distinct from the other papers from the project. Understanding the polarized nature (or lack thereof) of climate change beliefs/actions is essential and a unique contribution of this particular paper. I am satisfied with all the responses. My only suggestion concerns the abstract. I view the paper as making three distinct contributions: 1) demonstrating a green gap (a term the authors may want to also mention in the conclusion given its use in the first part of the paper), 2) showing that interventions are largely symmetrical when effective, and 3) showing that interventions to influence behavior are lacking (one success on just liberals suggests little success overall here-- given the backfire effects, even for liberals, no interventions may be preferable). Instead of detailing the precise interventions in the abstract, I suggest highlighting points 2 and/or 3 more explicitly. I think it will appeal to readers more than what is presently there. Of course, this is up to the authors and editor. I felt otherwise the paper reads well and recognizes limitations/contributions.

Signed (as requested by the journal),
Jamie Druckman

Reviewer #3 (Remarks to the Author):

The authors have addressed my previous concerns and suggestions.

REVIEWER COMMENTS:

Reviewer #2 (Remarks to the Author):

I appreciate the authors' attention to my comments and those of the other reviewers. I also do believe this paper is sufficiently distinct from the other papers from the project. Understanding the polarized nature (or lack thereof) of climate change beliefs/actions is essential and a unique contribution of this particular paper. I am satisfied with all the responses. My only suggestion concerns the abstract. I view the paper as making three distinct contributions: 1) demonstrating a green gap (a term the authors may want to also mention in the conclusion given its use in the first part of the paper), 2) showing that interventions are largely symmetrical when effective, and 3) showing that interventions to influence behavior are lacking (one success on just liberals suggests little success overall here-- given the backfire effects, even for liberals, no interventions may be preferable). Instead of detailing the precise interventions in the abstract, I suggest highlighting points 2 and/or 3 more explicitly. I think it will appeal to readers more than what is presently there. Of course, this is up to the authors and editor. I felt otherwise the paper reads well and recognizes limitations/contributions.

We thank the reviewer for their time and thoughtful feedback throughout this paper's revision process. We also appreciate the reviewer's positive assessment of our paper and its novelty and contribution!

In response to the minor text changes suggested by the reviewer, we have incorporated language in the abstract and discussion sections: In the discussion we added the following "Overall, the present work provides global evidence on the complex relationship between political ideology and climate change beliefs, policy support, and individual climate action, providing evidence for a conservative-oriented green gap." In the abstract, we now say that interventions are symmetrical: "We also find that three interventions (emphasizing effective collective actions, writing a letter to a future generation member, and writing a letter from the future self) boost climate beliefs and policy support across the ideological spectrum". We also added the following sentence to the abstract, in line with the reviewer's suggestion: "None of the interventions tested show evidence for a statistically significant boost in climate action for self-identified conservatives."

Reviewer #3 (Remarks to the Author):

The authors have addressed my previous concerns and suggestions.

We again thank the reviewer for their time and feedback for each round of revision.